



# Multi-scale spatialization of snow water equivalent (SWE) according to their spatial structures in eastern Canada

Noumonvi Yawu Sena[1], Karem Chokmani[1], Erwan Gloaguen[1]. Monique Bernier[1]

[1] *Institut national de la recherche scientifique Centre - Eau Terre Environnement 490, rue de la Couronne Québec (Québec) G1K 9A9 Canada*

*Correspondence to*: noumonvi_yawu.sena@ete.inrs.ca

**Abstract**

The spatial variability of snow plays a key role in snow water storage, spring runoff and hydraulic dam management. The snow survey network unequally distributed ability, to monitoring the spatial variability of the snow cover is limited. The spatial
variability of the snow cover is explained by physiographic factors, which generate spatial structures at different scales. The variability of the snow cover is explained by physiographic factors, which generate structures at different scales. These structures of spatial variability of the snow cover were delimited by a functional approach at the local (300 x 300 m) and regional (10 x 10 km) scales on eastern Canada. The territory was segmented into regions, (called spatial structures,) with homogeneous average maximum annual snow water equivalent (SWE).

The aim of this paper is to spatialize the average maximum annual snow water equivalent (SWE) according to spatial variability structures at both scales. Initially, at the regional scale, the average maximum annual SWE is estimated using the stepwise regression approach. Secondly, the SWE residuals are estimated using a regression approach on local physiographic meta-variables.

The estimated SWE allows quantifying the spatial variability of the average maximum annual SWE for regional and local
physiographic factors. Indeed, at the regional scale, the physiographic regional factors explain 68% of the variance of the spatial variability of the average maximum annual SWE. At the local scale, physiographic factors improve the estimate of the average annual maximum SWE by 21% (R = 89%) for an unexplained share of 10% of the variance. Local physiographic factors reorganize the regional residuals of average maximum annual SWE and contribute to the local variability. This study shows the role of altitude in snow accumulation at the regional scale, where the presence of high mountains increases the
amount of rainfall from wet winds. In each geographical area, the highest values of the SWE are related to high mountain peaks. The impact is confirmed at the foothills of the Canadian Shield mountains. At the local scale, the regional residual value was reorganized based on local physiographic factors (slope, forms of catchment, distance to rivers, etc.); this adjustment led to high SWE values in the concave landscape and the ubacs away from sunlight. The SWE accumulation area corresponds to the depressions and concave sections at foothills.

Keywords*: spatial variability, snow water equivalent, regression*



# 1. Introduction

Knowledge of the spatial variability of snow cover is very important because the snow cover plays an important role in many environmental aspects in Canada, e.g., water management, snow water storage, spring runoff and hydraulic dam management. Snow accumulations in Quebec and Labrador often exceed 300 mm in terms of snow water equivalent (SWE) (Brown, 2010). The main snow data are provided by the snow survey network, which was designed to monitor meteorological and hydrological forecasts in areas of economic interest (Brown et al., 2007; Brasnett, 1999; Braateen and Brown, 1998). Therefore, its density and spatial distribution are not optimal to analyze the spatial variability of the snow cover across the entire territory.

For example, in quantitative analysis spatial variability and the validation of spatial estimations made using remote sensing algorithms, major limit or obstacle is the uneven spatial distribution of the snow survey stations. Under this condition, the snow survey network's ability to reproduce the spatial variability of SWE is reduced (Sena et al., 2019). Because the number of snow sites is inadequate, some studies integrate data from other structures in the estimation process of their watershed study, regardless of the limitations and scale. In zones insufficiently covered by the snow survey, different approaches of spatial estimation such as the linear geostatistical methods (Erxleben et al., 2002), neural networks (Evora et al., 2008), and physical or hybrid models (Turcotte et al., 2006; Goovaerts, 2000) help to spatialize the physical parameters of snow cover (density, depth, SWE).

Focusing on SWE, Tapsoba, et al. (2005) used the geostatistical algorithm of kriging with external drift (KED) on in situ data applied to the Gatineau River watershed and used a digital elevation model with a resolution of 10 km as external drift. The KED approach shows a good estimations precision improvement notably in under-sampled and extrapolated areas. The authors of this study found that the integration of cofactors correlated with SWE could improve the accuracy of these estimates. At the regional scale, Watson et al. (2008) explain how, orographic mountain ranges and valleys influence the overall distribution of precipitations and, hence, the snow water equivalent. Other factors, such as solar radiation, vegetation cover, and exposure to high winds act to reduce snow pack accumulations at highly localized scales. Elder et al. (1991) evaluated a stratified sampling scheme by identifying and mapping the zones of similar snow properties, based on the topographic parameters (elevation, slope and radiation values) that induce accumulation and ablation variations in the heterogeneous landscape of the Madison headwaters system of the central Yellowstone National Park. Carroll and Cressie (1997) estimated the SWE in the North Fork Clearwater River watershed by using a positive-definite spatial covariance function that incorporated geomorphic site attributes when SWE estimates were obtained. Remote sensing (optical, passive or active microwave, etc.) is also used as an alternative source of indirect snow cover data. It provides results on the presence or absence of snow (De Sève et al., 2008; Chokmani et al., 2005; Chokmani et al., 2006; Brown and Goodison, 1996).

Factors such as snow conditions (deep snow cover) and vegetation cover (tundra, taiga, boreal forest, agricultural area) represent a challenge to the development of an efficient approach to estimate the physical properties of snow and to measure its spatial variability (Turcotte et al., 2006; Goïta et al., 2003; Brown, 2000).





The study of multi-scale analysis of the spatial variability of the SWE in eastern Canada by Sena et al (2015), has applied a functional approach under the assumption that the snow event was stationary over the entire observation period (Sena et al., 2015). First, the different spatial variability structures of the mean annual maximum of the SWE were visually identified, according to their geographical positions (latitude and longitude). According on the similarity of the values collected at each

station, they were quantitatively selected using the spatial association index. The two methods of this study show that the SWE was not stationary neither in terms of its mean nor its variance across the study area. This study demonstrated the spatial structure limits remain subjective. In this case, the spatial segmentation approach is justified for delimiter explicitly the spatial structures. Subsequently, the spatial segmentation algorithm was used to elicit delineate the boundaries of the SWE spatial variability structures. Sena et al. (2015) applied a Canonical Correlation Analysis (CCA) to analyze the spatial variability of

snow. This allowed to identify the spatial structures based on the physiographic meta-variables obtained by the CCA. The CCA is applied to regional physiographic factors (latitude, longitude, altitude and distance to ocean) and local physiographic factors (slope, aspect, distance to rivers, solar radiation, curvature, and vegetation height) and average maximum annual SWE (Séna, 2015). This physiographic factor holds in place the underlying processes that generate the homogeneous areas (called SWE spatial structures), that have a strong contrast with their surrounding area according to the different observation scales.

Canonical correlation analysis has allowed to obtain physiographic metavariables that are integrated in the spatial segmentation algorithm. Finally, the spatial segmentation results were validated by comparing snow data from the adjacent geographical zones using the Kruskal–Wallis nonparametric statistical test (Séna et al.2015). As a result, the structures identified are different from another and the physical spatialization parameters of snow cover (such as snow water equivalent) must take into account the physical boundaries to reduce errors and biases in the spatial estimation (Séna, 2015).

To respond to this need, this study proposes to spatialize the SWE according to the structures of spatial variability of SWE. The main objective of this study is to develop a multi-scale spatialization approach by taking into account the structures delineated in the spatial variability analysis of the SWE at both scales (local and regional) by Sena et al. (2015). The objectives are the following:

   1 - estimate the SWE according to the physiographic meta-variables delimited by spatial structures at a regional scale

25   (10 km)

   2 - spatialize the SWE at the local scale (300 m), based on SWE residuals and local physiographic meta-variables. The first part addresses the regional scale, and a polynomial multi-function is applied to the physiographic meta-variables and average maximum annual SWE recorded at the snow survey stations included in the study area. Kriging was applied to the residuals of each spatial structure obtained by a variogram analysis to improve the SWE estimation. In the second part,

30   addressing the local scale, a regression approach is applied to the local physiographic meta-variables and the residuals in each unit area. The SWE values estimated at the local scale correspond to the regional estimation adjusted by the residuals obtained from local physiographic meta-variables.



## 2. Material and methods

### 2.1 Study site and description of SWE structures

The study covers the province of Quebec and Labrador region, extending from 45°–65 °N to 55°–82 °W (Fig. 1). The relief is moderate and includes three sets (MDDELCC, 2001): Appalachia, St. Lawrence Lowlands and the Canadian Shield.

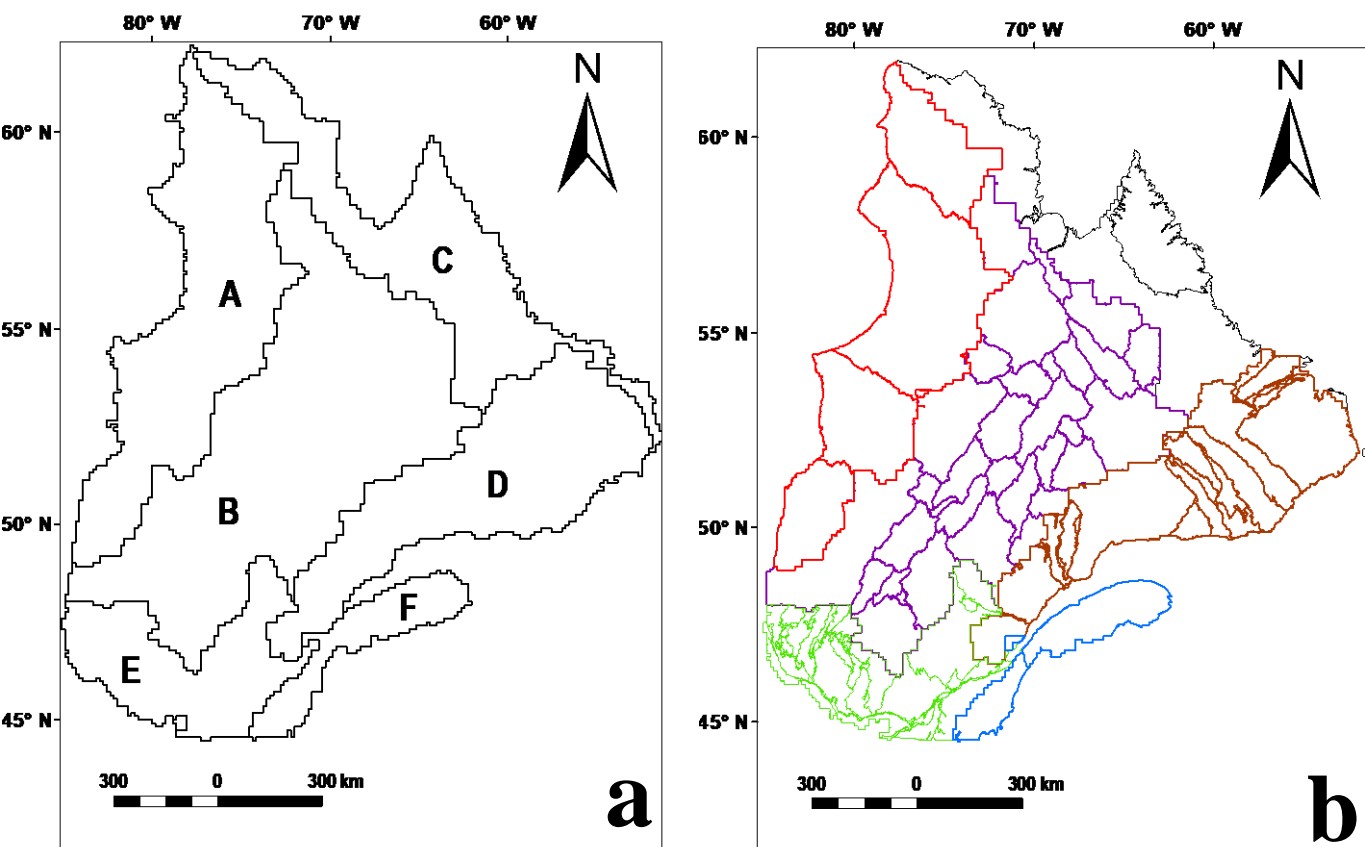

Figure 1: Geographical areas with homogeneous SWE spatial structures and the delimited zones at the regional (a) and local (b) scale obtained with the functional approach (Sena, 2015)





Séna et al. (2015) applied the multi-resolution spatial segmentation algorithm which led to the identification of six geographical areas with homogeneous SWE spatial structures at the regional (10 km x 10 km) and local scales (300 m x 300 m) (Fig. 1a).

At the regional scale, the multi-resolution spatial segmentation algorithm led to the identification of six geographical zones with homogeneous SWE spatial structures. The identification process of the structures was based on physiographic factors, geographic position (latitude and longitude), relief, and distance from the ocean. SWE spatial structures can be considered similar to the layout of landforms (altitude) and climate classes. A description of the main structures used in this study is provided below.

The eastern zone (A) of the study site includes the plains bays (Hudson and James) and the foothills located east of the high reliefs of the Canadian Shield. The central zone (B), located in the northeast-southwest axis included much of the medium relief (500-700 m) and mountains (Tichégami 776 m, Otish 1128 m). North of the study area, the hilly landscape of the high lands of northwestern (Pingualuit Crater) and northeastern Labrador (Jacques-Rousseau Mountain 1261 m) are grouped in zone C. Along the St. Lawrence River is zone D, which includes the highlands of northern and northeastern Labrador and the eastern part of the Canadian Shield. The maritime part of the study area, including the Appalachian Mountains of Gaspésie and the St. Lawrence River, are grouped in zone F. In south, the lowland of the Laurentians, the Outaouais and Abitibi-Témiscamingue are grouped in zone E.

At the local scale (300 m x 300 m), the multi-resolution spatial segmentation algorithm allowed the limit of several geographical areas. The identified local structures of SWE spatial variability demonstrate the dominant role of the physiographic factors (slope, curvature, orientation, distance to rivers etc.) (Fig. 1b) in the maintenance and redistribution of snow cover. At this scale, geographical areas with homogeneous SWE spatial structures were identified based on local underlying processes controlled by physiographic factors such as slopes, curvature of slopes, aspect, solar radiation, distance to rivers, plant height. The local characteristics of watershed on each zone contribute to the variation of snow accumulations (Sena et al., 2015). The spatial structures of the SWE used in this study thus varied depending on the observation scale.

## 2.2 Snow data

The measurement site must be representative of the variability of the surrounding snow cover and topography (MSC, 2004). At each measurement site, snow depth was measured. A core sample was extracted, which allowed to evaluate the SWE and density of the snow pack. At each marker where the snow depth was greater than 25 cm, the weight of the core sample was measured. The SWE was obtained by subtracting the weight of the empty snow sampler from the total weight of the sample. On the other hand, if the snow depth was less than 25 cm at one or more of the markers, the SWE was obtained after subtracting the weight of the empty snow sample bucket from the cumulative measurement of the cores of all markers. Values representative of the entire sampling site were obtained by calculating the mean of the 10 sampling sites. Measurements were carried out biweekly and were generally limited to the period ranging from January to May (MSC, 2004).

A ten-year observation period is the minimum necessary to cover cyclical atmospheric and oceanic events (solar cycle, El Niño-Southern oscillation, La Niña, etc.) that can influence the variability of the snow cover (Brown, 2000; Rasmussen et al., 1999; Sobolowski and Frei, 2007). Based on the hypothesis that the snow phenomenon was stationary during a ten-year observation period, the sub-periods were not considered. Therefore, out of the 426 snow survey stations constituting the network of the studied territory, we considered the 367 stations for which historical data was available for at least the last 10 years. The data from these stations was provided by our partners, including data from 193 stations of the province of Quebec's Ministère du Développement Durable, Environnement et Lutte contre les Changements Climatiques (MSC, 2004), 19 stations belonging to RioTinto, 76 to Hydro-Québec, and 79 provided to Environment and Climate Change Canada. For each retained station, the mean annual



maximum SWE was calculated. Thereafter, statistical descriptors of central tendency (mean of annual maxima) and dispersion (standard deviation, interquartile interval, median) of the mean annual maximum SWE were calculated.

## 2.3 Method

The multi-scale spatialization approach based on spatial variability structures of SWE is shown in Fig. 2.

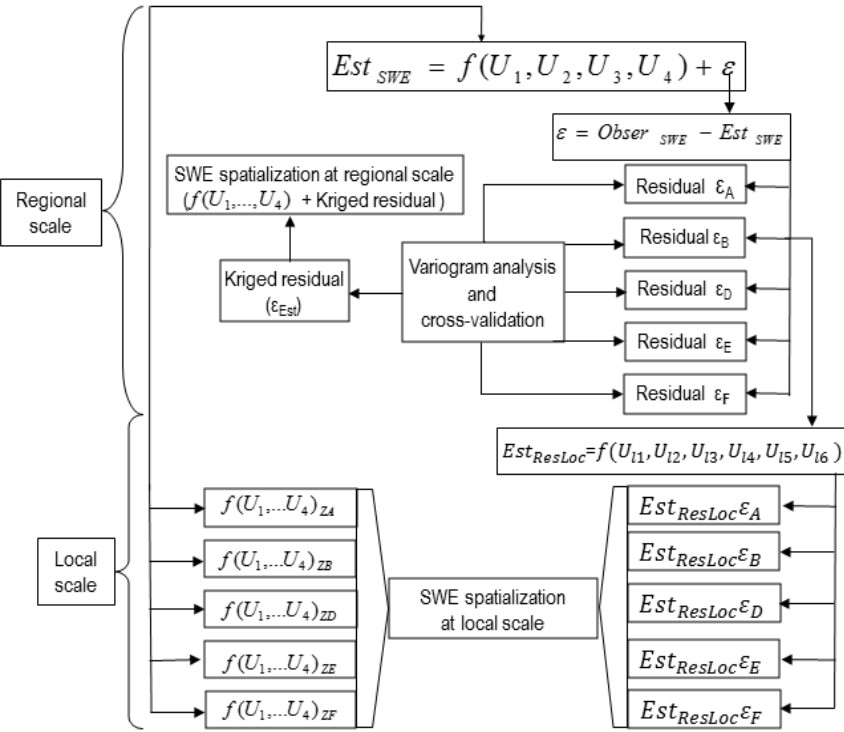

Figure 2: Diagram of the SWE spatialization methodological approach applied to the spatial structures defined at the regional and local scale.

## 2.4 Stepwise regression approach

This study proposes to spatialize the SWE according to the different structures of delimited spatial variability at regional (10 km x 10 km) and local (300 m x 300 m) scales, as presented by Sena et al. (2015).

### 2.4.1 Regional scale

The stepwise regression is a linear regression in which the relationship between the independent variable X (physiographic meta-variables) and the dependent variable Y (SWE) is modelled as an $n^{th}$ degree polynomial equation (Borcard, 2005). At the regional scale (10 km x 10 km), the physiographic metavariables ($U_1, U_2, U_3, U_4$), and all the collected average maximum annual SWE data were used in the stepwise regression. In statistic, the choice of predictive variables carried out by an automatic procedure, is one of an approach of stepwise regression model fitting. According some pre- criterion, a variable is added or subtracted from the set of explanatory variables (Draper and Smith, 1998). The backward elimination is the main approach adopted. First, all





candidate variables are involved and testing the deletion of each variable basing a chosen model fit criterion. Next, deleted the variable whose loss gives the most statistically insignificant deterioration of the model fit. Finally, the process is repeated until no other variable can be removed without a statistically insignificant loss of adjustment. In this study, the MATLAB R2014a statistical analysis tool interrupts the operation when all variables not in the model have a p-value greater than the specified Alpha value for inclusion and all variables in the model have a p-value less than or equal to the specified Alpha value for exclusion.

The stepwise function adjusts the explanatory variables to the dependent variable in a non-linear manner and involves adding the newly obtained predictor in squared, cubed, etc. The prediction function is according to $f_{(x)}$ in Eq. (1):

$$f(x) = a_0 + a_1 x_1 + a_2 x_2 + \cdots + a_n x^2 + a_{n+1} x_2^2 + \cdots + a_{nm} x_n^m \qquad (1)$$

Where $a_0$ and $a_n$ are the adjustment coefficients of the prediction model and $\mathcal{X}$ the secondary variable defined on X element in $R^2$.

The combination of the trend and the residuals results in the random $SWE_{Est}$ function (Eq. 2):

$$SWE_{Est} = f_{(x)} + \varepsilon \qquad (2)$$

Where $\varepsilon$ is random (residual) with a null mean and the variogram describes the spatial dependence at a small scale.

At the regional scale, the residuals $\varepsilon$ ($\varepsilon = SWE_{Obser} - SWE_{Est}$) were extracted for each homogeneous SWE zone. These estimated residuals were added to estimate the SWE. The variogram analysis consists of the experimental variogram calculated from the data (residuals) and the variogram model fitted to the residuals data (Goovaerts, 1997). The variogram is defined as the variance of the difference between field values at two locations across realizations of the residuals field. The residuals with a spatial structure underwent kriging and were retained to improve the deterministic part of the estimated SWE (Fig. 2). Ordinary kriging was performed to interpolate the residual of the regression model resulting from the variogram analysis (further information can be found in Goovaerts (1997)). The variogram indices ($C_0$ = nugget effect and C = variance) are used to characterize the spatial structure of residuals by considering the following ratio (β) (Eq. 3):

$$\beta = \frac{C}{(C + C_0)} \qquad (3)$$

-if β has a high value, the nugget effect $C_0$ is very small ($< 0.5$) and the spatial variability is due to the distance between the snow survey stations; $C_0 < 10\% C$;

-if β has a small value, the nugget effect $C_0$ is high ($> 0.5$) and the spatial variability of residual is due to the nugget effect.

### 2.3.2 Local scale

At the local scale, the residuals of the regional estimation of the SWE are reorganized by local physiographic factors to develop specific snow accumulation estimations for each zone. To do that, the residuals are estimated based on local physiographic metavariables. Thereafter, local SWE estimates correspond to the regional estimate corrected by the residuals calculated using the local physiographic metavariables of each area (Fig. 2). For this purpose, the stepwise regression model exploits local physiographic metavariables and residuals from the regional estimate of each area. The models are verified by cross validation methods (further information can be found in Polikar (2006)). Four statistical evaluation indices were used (determination coefficient (R2), BIAIS, relative mean squared error (RMSE), and Nash–Sutcliffe efficiency) and they are presented in Table 1. In zone C, the small number of snow survey stations (3) did not allow to proceed with the stepwise regression. The resampling process is applied to the SWE estimated at the regional scale (10 km x 10 km) to have SWE estimates at the local scale (300 m x300 m).





**Table1:** Statistical evaluation indices

$$R^2 = \left[ \frac{\sum_{i=1}^{n}(M_i - \overline{M})(E_S - \overline{E}_S)}{\sqrt{\sum_{i=1}^{n}(M_i - \overline{M})^2} \sqrt{\sum_{1=1}^{n}(E_s - \overline{E}_s)^2}} \right]^2 \qquad (4)$$

$$BIAIS_r = \frac{1}{n} \sum_{i=1}^{n} \left[ \frac{E_{si} - M_i}{M_i} \right] \qquad (5)$$

$$EQM_r = \sqrt{\frac{1}{n} \sum_{i=1}^{n} \left[ \frac{E_{si} - M_i}{M_i} \right]^2} \qquad (6)$$

$$NASH = 1 - \frac{\sum_{i=1}^{n}(M_i - E_{si})^2}{\sum_{i=1}^{n}(M_i - \overline{M})^2} \qquad (7)$$

where n is the sample number; M and $E_s$ SWE data measured and estimated; $\overline{M}$ and $\overline{E}_s$ mean values of SWE data measured and estimated.

### 2.5 Mapping approach for SWE

SWE spatialization maps were modelled using PCI Geomatica 2012 and ArcGis 9.0 softwares for both observation scales (regional and local) was modelled using PCI Geomatica and ArcGis software. At the regional scale, SWE mapping was achieved by combining the information contained in the structures delimited by according to the delimited structure in projects composites to the physiographic metavariables ($U_1, U_2, U_3, U_4$), and the residues having a spatial structure. The structures were kriged using a standard kriging approach. At the local scale, SWE mapping was performed according to the SWE estimated at the regional scale and the estimated regression residuals based on the physiographic metavariables ($U_{1LZ}, U_{2LZ}, U_{3LZ}, U_{4LZ}, U_{5LZ}, U_{6LZ}$).

## 3. Results

### 3.1.1 Stepwise regression approach based on the structures obtained at the regional scale

At the regional scale, the stepwise function is based on the physiographic meta-variables $U_1, U_2, U_3$ and $U_4$ and SWE dataset, as presented in Eq. 4.

$$EstSWE_{ER} = 45.2U_1 + 8.4U_4 - 19.2U_2^2 + 11.8U_3^2 + 3U_3^3 + 2410.8 \qquad (4)$$

The variogram analysis was not applied to the residual of zones A and C because the number of snow survey stations was too low (8 and 3, respectively). To alleviate the text, the variogram of zone E is presented in Fig. 3, but the variogram of the other residuals (B, D and F) are available in Figure A 1.

In zone B, the variogram analysis of the SWE residual has a variance ($\gamma_{(h_B)}$) of 1100, with an increment of 10 km over a distance of 42 km (Figure A 1, residue variogram of zone B). With a nugget effect of 260, the ratio (β = 0.8) shows a high spatial variation due to the distance between the snow survey stations. Based on this we can confirm that the residual structure is due to the distance between sites.

The same ratio ($\beta = 0.8$) is observed in zone D, illustrating that the SWE residue variation is also due to the distance between stations rather than the nugget effect. The nugget effect ($C_0 = 1000$) of zone D is the highest observed in this study (Figure A 1 D). The variance between stations ($\gamma_{(h_D)} = 4200$) is observed at 100 km for increment of 2 km (Figure A 1, residual variogram of zone F). The nugget effect ($C_0 = 90$) observed at the regional scale of zone E was the smallest one among delineated zones.

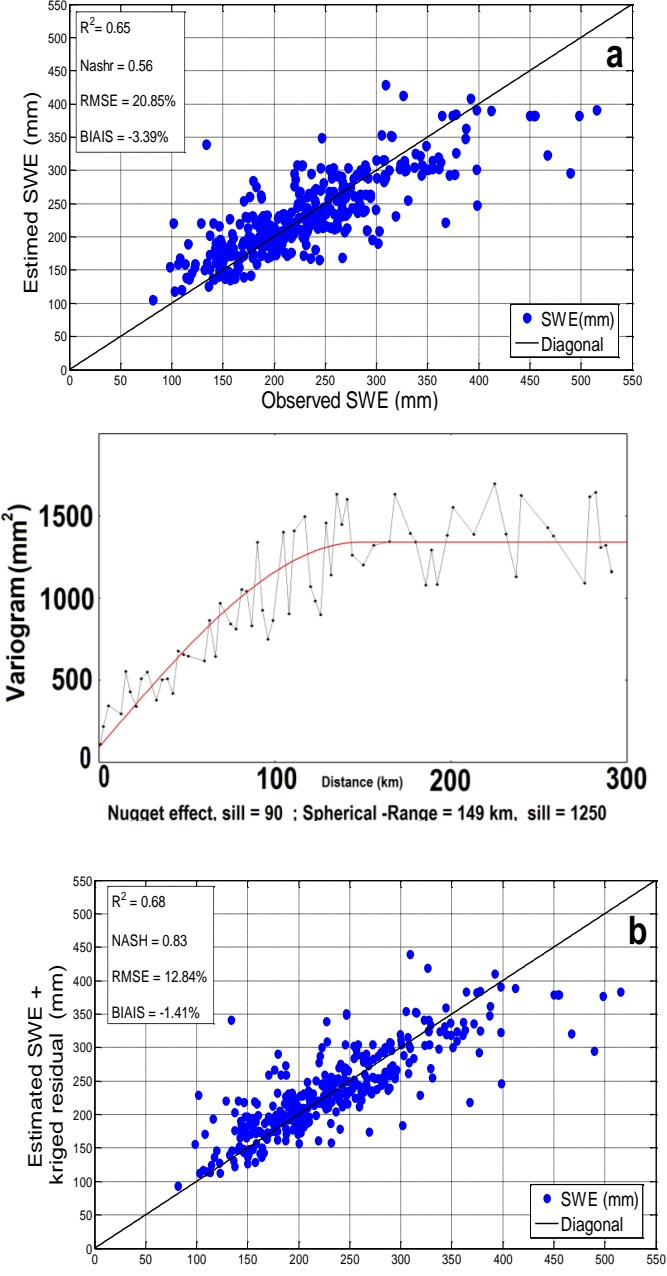

Figure 3: SWE data observed versus data estimated by the model (a), the model adjusted by residues (b) with model performance indices and the variogram model applied to estimate residues throughout the study area (top right)





This can be explained by the high sampling density, which reduces the variations due to the distance between stations, which is inferior to the increment step. Furthermore, at a distance of 150 km, the correlation between the residuals of the stations is high, with $\gamma_{(h_E)} = 1250$ (variance of the SWE residues in zone E).

In zone F, the SWE residual variogram has a high value; with $C_0 = 510$, and a variance of $\gamma_{(h_F)} = 400$. A low ratio ($\beta = 0.4$) shows that the variation is due to the nugget effect. This value indicates a partial absence of correlation between the SWE values observed at two very close stations. The nugget effect is explained by a low resemblance between the regionalized values taken in close proximity.

### 3.1.2   Validation approach at regional scale

The model estimated SWE values and their variance by using the determination coefficient $R^2 = 0.65$. This value shows that 65% of the variation of the average annual maximum of SWE is explained by the regression model and 35% remain unexplained. The estimates show that the model tends to underestimate the highest SWE values (BIAIS = -3.39%) with reference to the diagonal line (Fig. 3a). The variogram analysis of the SWE tailings was carried out in zones B, D, E and F (see Figure A 1). In zone A, the number of stations (18) is very small, which is a limitation to the variogram analysis. In zone

B, the variogram analysis of the SWE residuals shows a variance of 1100 mm$^2$ up to the distance of 42 km at 10 km intervals (Figure A 1B). With a nugget effect of 260 mm$^2$, the ratio ($\beta = 0.8$) shows that the spatial variation is due to the distance between stations. However, the estimation of SWE residuals as a function of the regional metavariables evaluated by the cross-validation shows points that are very far from the 1:1 diagonal (Figure A 3B).

   In zone D, the nugget effect of the SWE residue ($C_0=1000$) is the highest of all zones. The variance between stations (C=

4200 mm$^2$) is reached at 100 km with a 2 km interval (Figure A 1D). The ratio ($\beta = 0.8$) also shows that the variation is due to the distance between stations and not to the nugget effect. In zone D, SWE residuals explain 57% of the variance of the SWE residuals for 43% unexplained (Figure A 3D). The same is observed in the case of zone B, where the point distribution in the plot of SWE residuals versus SWE estimated by the variogram model (see Figure A 3 D) is also far from the 1:1 diagonal (Figure A 3D). In zone D, the estimated SWE residual explains 57% of the variance and the balance remains unexplained.

Moreover, the residual's distribution does not fit with the diagonal (Figure A 1D).

   In zone E, the variogram analysis of the SWE residues has a nugget effect of 90 mm$^2$. This is the lowest nugget effect observed through all zones. This can be explained by the high sampling density, which attenuates variation at distances inferior to the increment step. On the other hand, at a distance of about 150 km, the correlation between the residuals of the SWE stations is strong and reflected by the spatial structure C=1250 mm$^2$, (Fig. 3C). The SWE residuals obtained with the spherical variogram

model explains 70% of the variance of the residuals and 30% remain unexplained (Figure A 3E). The ratio ($\beta = 0.9$) shows that the variability of the SWE in zone B is mostly explained by the distance. On the other hand, the graph of the measured SWE residuals estimated versus those estimated by the variographic model shows a good distribution of points with respect to the 1:1 diagonal.





In zone F, the variogram analysis of the SWE residues shows a variance of 800 mm² and a nugget effect (microvariation) of $C_0 = 510$ mm ² (Figure A 1F). The ratio ($\beta = 0.4$) is very low, indicating that the variation is due to the nugget effect. Also, the results of the cross validation show that the SWE distribution is far from the 1:1 diagonal. The variance of the SWE residuals explained by the variographic model is near zero. The SWE variance of zone F, as explained by the residual variogram model,

is null (Figure A 3F).

The cross validation to predict the effectiveness of variogram models on residuals shows overall that the variogram model on the zone E provides a better result (R= 0.70). For this purpose, only the estimated residual in zone E was added to the regional estimate to adjust the estimated value of the SWE at the regional scale. The results of the adjusted model by the residuals of the zone E improved the variance by 3% ($R^2 = 0.68$) and 32% remain unexplained. The Nash–Sutcliffe efficiency test

indicates that the estimated residual model is robust with a success rate of 83% (Fig.3b). The RMSE and BIAIS were reduced by 2% and 8%, respectively, compared to models without adjustments. The robustness of the model is also confirmed by comparing the measured SWE with the estimated SWE, which shows a better distribution of the points along the diagonal.

### 3.1.3 Stepwise regression approach at the local scale

As noted above, stepwise regression models were applied to the regional SWE residuals according to the local metavariables

observed in each of the six zones. This does in fact make it possible to estimate the SWE by using local variables (Fig. 2. Diagram of the SWE spatialization methodological approach applied to the spatial structures defined at the regional and local scale). The stepwise regression models of the SWE residuals at the local scale are presented in Eq. 5-9:

$$Est_{ResLoc}\varepsilon_A = -19.1U_{1LZA} - 5.1 \tag{5}$$

$$Est_{ResLoc}\varepsilon_B = 9.4U_{4LZB}^2 + 4.1 \tag{6}$$

$$Est_{ResLoc}\varepsilon_D = 17.6U_{1LZD}^2 + 6.1U_{1LZD}^3 - 14.2 \tag{7}$$

$$Est_{ResLoc}\varepsilon_E = -6.3U_{2LZE} - 7.1U_{3LZE} - 5.2U_{1LZE}^3 + 0.7 \tag{8}$$

$$Est_{ResLoc}\varepsilon_F = -22.7U_{1LZF} + 5.1U_{1LZF}^3 + 5.8U_{1LZF}^3 - 1.8 \tag{9}$$

Where $U_{LZ}$ is the local physiographic metavariables of each zone (A, B, D, E and F).

### 3.1.4 Validation approach at the local scale

Figures 4a and 5a demonstrate that local regression models of residuals as a function of local physiographic metavariables in zones A and B are not very strong. In zones A and B, the SWE residual estimation model takes into account the physiographic metavariable $U_{1LZA}$ and $U_{4LZB}$. Indeed, local residual models explain 35% and 30% of the variation in residuals in zones A and B respectively (Fig. 4a and 5a). The Nash values (-0.77 and -0.38) indicate that the models are not robust in the restitution of the regional residuals of SWE observed. These negative Nash values indicate that the mean residual values are considered

more precise than the results obtained by the modeling. The performance of the residual regression models can be assessed by



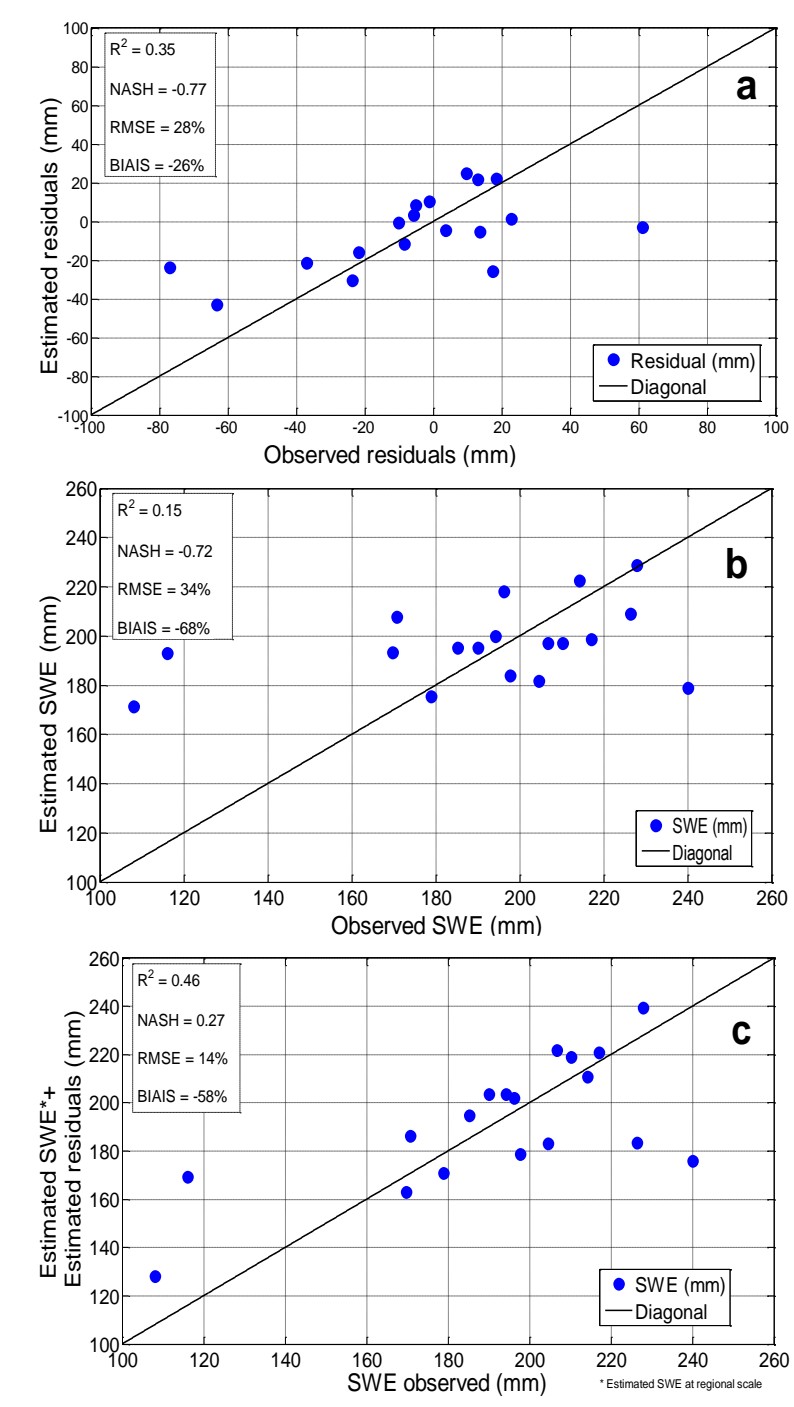

Figure 4: In zone A at local scale: (a) Estimated residuals of the SWE by the zonal model- (b) Observed values versus regional estimate of the SWE - (c) Observed values adjusted by the residuals estimated by the zonal model for the SWE in (a)





observing the distribution of points in relation to the 1:1 diagonal of the plots. The dispersion of the points is greater for extreme residual values (e.g., Fig. 5a). This suggests that the majority of the error calculated (RMSE) for all models would be affected by the extreme values. The negative BIAIS (-26% and -81%) indicates that the models tend to underestimate the residuals. Figures 4c and 5c shows the adjustment of the SWE estimates based on the residuals in relation to the local regression model

of the zones A and B. The local model explains 35% (Fig. 4a) of residual variance as a function of the local physiographic metavariables and increases the variance by 31% relative to the regional estimates ($R^2 = 0.46$ versus 0.15) (Figs. 4b and 4c). The Nash value of the local estimate is very low (0.27 in Fig. 4c), but it is better than the one observed at a local non-fitting model with a negative Nash (-0.72).

The same pattern is observed in zone B (Fig. 5c), where the addition of the estimated residuals explains 74% of the variance

of the local SWE and improves the variance by 16% (Fig. 5b and 5c). In this case, the SWE estimation model performs better with a higher Nash (0.65). The sum of the residuals estimated with the local metavariables reduces the BIAIS and RMSE values by 19% and 3% respectively (Fig. 5c).

In zone D, the SWE residual estimation model takes into account the physiographic metavariable $U_{1LZD}$. The result of the cross-validation shows that the local regression model of residuals as a function of local metavariables is not very robust (Fig. 6a).

It explains only 21% of the variance of the residuals of the SWE. Furthermore, the negative Nash value (-0.23) also indicates that the mean values of zone D residuals are more accurate than the results obtained by modelling. The dispersion of the points along the 1:1 diagonal shows the low robustness of the model with an RMSE of 74%. Adding the estimated residuals to the regional estimate slightly improves the local variance of the SWE, which is of 4% (Fig. 6c). The validation indices (Nash = 0.79 and RMSE = 12%) show that the adjustment model performs slightly better than the regional estimate when

compared to the observed SWE data (Fig. 6b).

The models used to estimate the SWE residues in zones E and F as a function of local physiographic metavariables take into account $U_{1LZE}$, $U_{2LZE}$, $U_{3LZE}$ (Eq. 8) and $U_{1LZF}$ (Eq. 7) respectively. In these zones, the Nash value is negative. In zone E, the residuals model explains 35% (Fig. 7a) of the variance compared to 23% in zone F (Fig. 8a). In addition, the points along the 1:1 diagonal in zone F are less scattered than in zone E. The negative value of BIAIS in both zones shows that the model used

for residual estimations tends to underestimate SWE values. In zone E, the local SWE estimation results perform the best, with a higher portion of variance being explained by the residuals model (85% in zone E and 64% in zone F) and a higher Nash value than in zone F (0.83 in zone E and 0.70 in zone F) (Fig. 7c). The dispersion of the points along the 1:1 diagonal is more clustered with an RMSE of 10.3% compared to the 18% observed for zone F.



Figure 5 : In zone B at local scale: (a) Estimated residuals of the SWE by the zonal model- (b) Observed values versus regional estimate of the SWE - (c) Observed values adjusted by the residuals estimated by the zonal model for the SWE in (a)





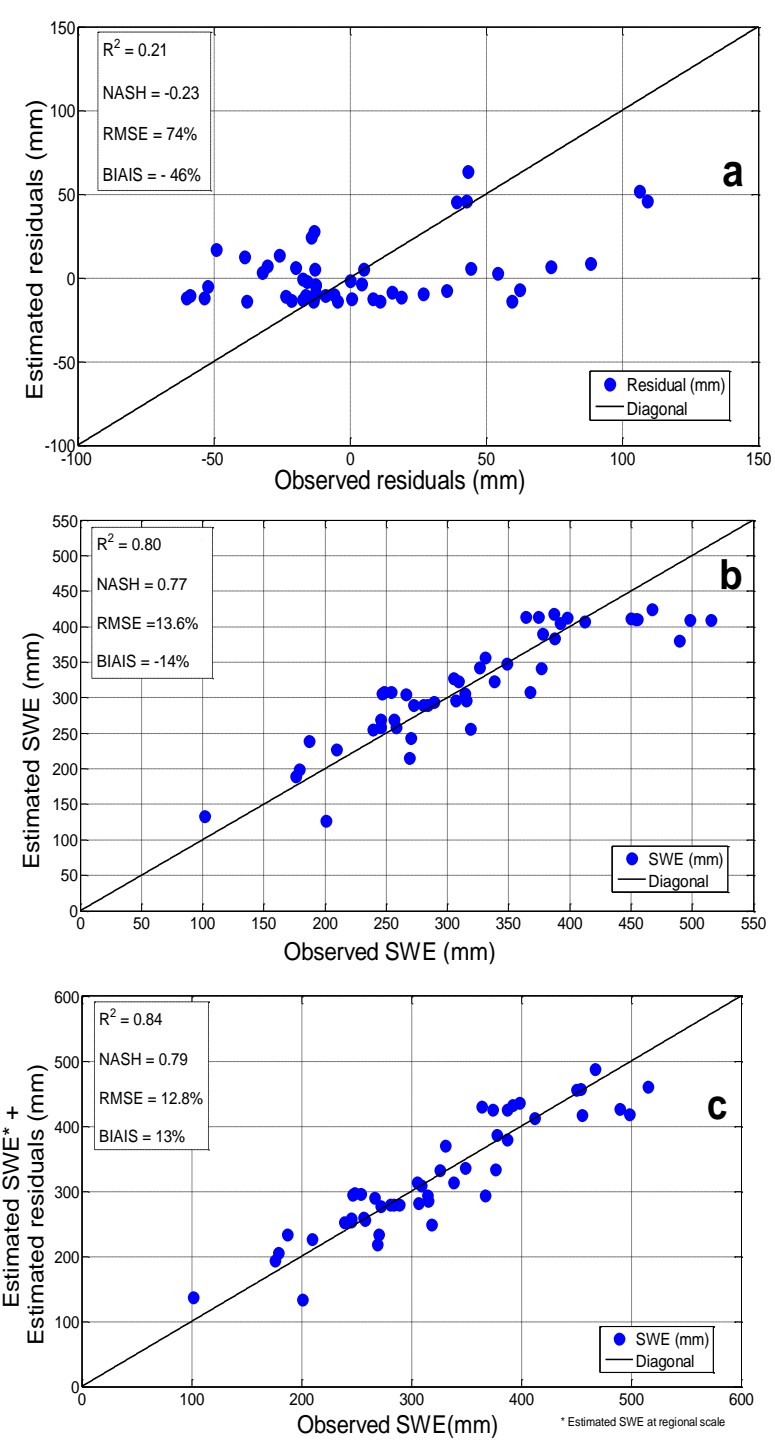

Figure 6 : In zone D at local scale: (a) Estimated residuals of the SWE by the zonal model- (b) Observed values versus regional estimate of the SWE - (c) Observed values adjusted by the residuals estimated by the zonal model for the SWE in (a)





Figure 7 : In zone E at local scale: (a) Estimated residuals of the SWE by the zonal model- (b) Observed values versus regional estimate of the SWE - (c) Observed values adjusted by the residuals estimated by the zonal model for the SWE in (a)





Figure 8 : In zone F at local scale: (a) Estimated residuals of the SWE by the zonal model- (b) Observed values versus regional estimate of the SWE - (c) Observed values adjusted by the residuals estimated by the zonal model for the SWE in (a)





Over the whole study area, at the regional scale, the linear relationship between the regional variables (distance to the ocean, altitude, latitude and longitude) and the SWE shows a weaker relationship ($R^2 = 0.68$) (Fig. 3). At the local scale, the SWE adjusted by the estimated residuals according to local metavariables (slope, orientation of slopes, land cover, solar radiation, distance to lakes, curvature and height of vegetation) shows a strong linear relationship ($R^2 = 0.89$) (Fig. 9).

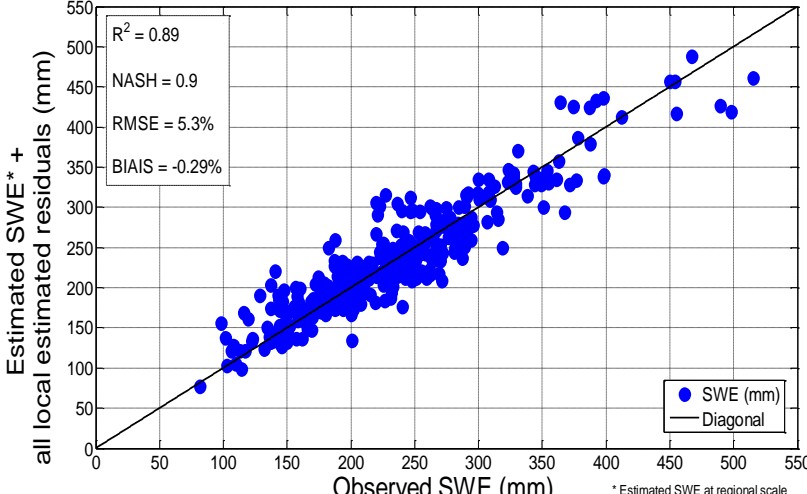

Figure 9: Observed SWE values versus all local SWE adjusted by the residuals estimated based on local physiographic metavariables. Their performance indicators are indicated at the local scale.

The linear relationship increases by 21% relative to the regional scale and that can be explained by the role played by local variables in snow accumulations. At this scale of observation, snow cover accumulation is controlled and guided by sets of subprocesses controlled by these local variables. With a very high Nash indicator (0.9), the linear relationship between the sum of estimated SWE residuals based on physiographic metavariables and observed SWE data shows that the model is very efficient in estimating SWE values.

## 3.2 SWE mapping according to the delimited structures

### 3.2.1 Regional scale

At the regional scale (10 km x 10 km), the SWE spatialization was performed in each of the delimited structure. In zone A, along the foothills of the Canadian Shield, SWE values are of 250-300 mm (Fig. 10A). North of the Inukjuak snow survey station, the D'Youville and Puvirnituq mountains are also very snowy, with a value of 250-300 mm all the way to Hudson Bay (Fig. 10A). The snow survey stations located at Bolem, Kanaapscow, Bienvielle and Mollet lakes on the foothills of the Canadian Shield and at Qingaaluk (400 m) and Umiujaq (400 m) hills also have a high SWE value (250-300 mm). The value gradually decreases west of the foothills of the Canadian Shield towards the bay's lowlands. This shows that in this area, high altitude constitutes a barrier for the wet winds, resulting in more snow on the western foothills and along the coast.







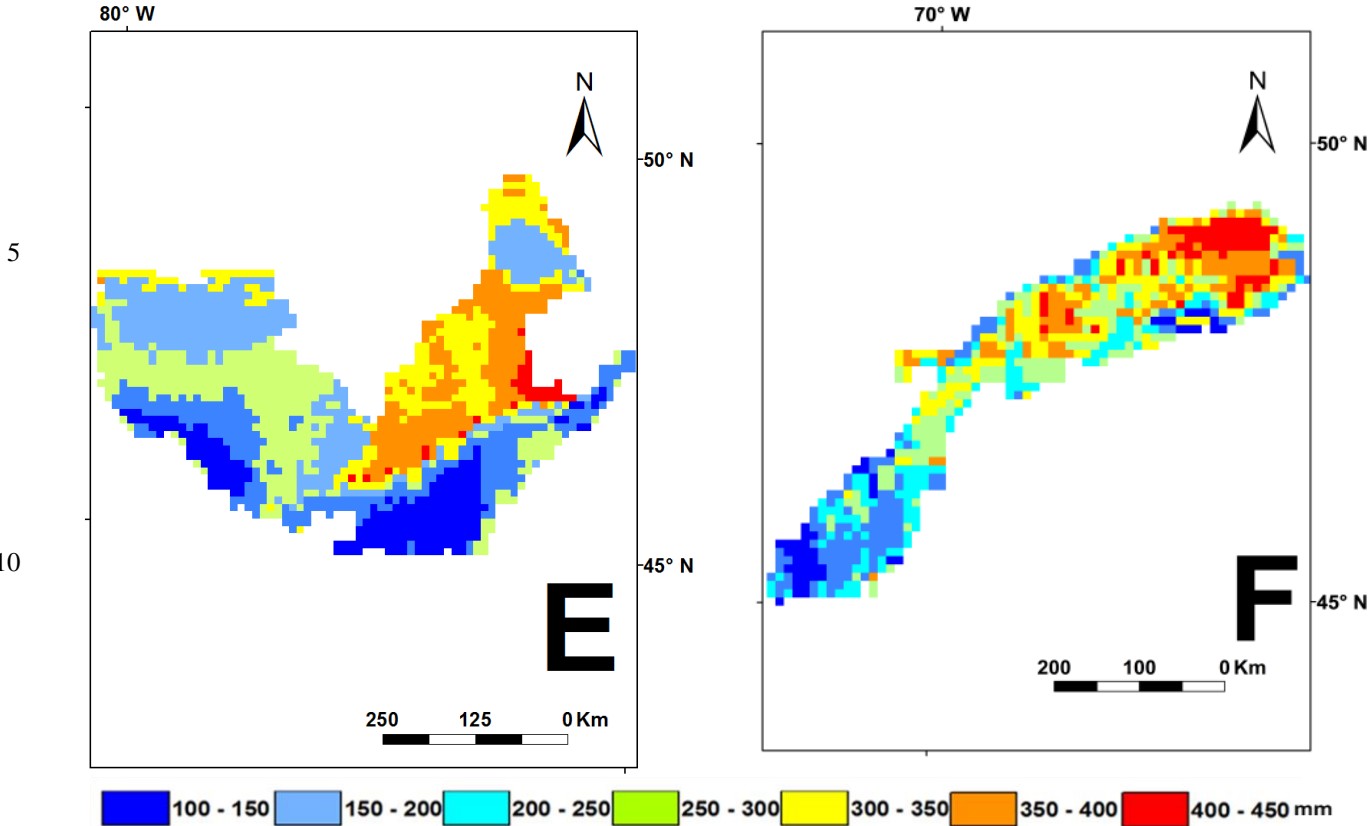

Figure 10: Average maximum annual SWE on the spatial structures delimited by the functional approach at the regional scale in eastern Canada.

In zone B, the SWE ranges from 200 mm to 300 mm on Tichégami, Otish Témiscamie, Severson and Uapahtkueh mountain line (Fig. 10B). The eastern part of the mountain foothills has an estimated SWE between 300 and 400 mm. The foothills of Mount Otish presents the SWE values between 300 and 350 mm, and 350 and 400 mm in concave section. The same value is also estimated on the great peaks of the Laurentian Mountains (Mount Kaoskiwonatinak (627 m). In the southwestern part of zone B, the estimated SWE is between 300 and 350 mm on the high reliefs of the Laurentians (Mont Reid, 424 m) and 250-300 mm in the southwest foothills of Témiscamie Mountains. To the north, a high precipitation area (300-400 mm) that acts as a barrier to polar winds for the northernmost mountains (Jacques-Rousseau Mountains Iberville). In sum, the highest values are found on the Canadian Shield (province of Greenville and Lake Superior).

In zone C (Fig. 10C), the SWE spatialization shows values between 300 mm and 350 mm over all northern limits with high peaks (400 m) of Torngat Mountains. Indeed, the northern limit of the study area is mountainous and includes Labrador Sea, Torngat Mountains (Jacques-Rousseau 1261 m, Iberville, 1662 m) and the Pingualuit crater along the Hudson Bay Strait to the north. These mountains act as polar wind barriers, leading to high snow accumulations on the north side. The lowlands of Ungava Bay have SWE values between 250 mm and 300 mm. These mountains extend to the east and constitute the highest





peaks in zone D. As in zone C, north of zone D, the high values (300-400 mm) are found on the northern mountains (from Happy Valley Goose Bay to Churchill Falls). The same value is found along the Laurentian peaks and along the St. Lawrence River. South of zone D, the high values illustrate the role played by altitude with regard to snow accumulations, at the Grands-Jardins national park (Raoul Blanchard (1175 m), Erables (1033 m), etc.) and Valin mountains (984 m) (Fig. 10 D (Average maximum annual SWE on the spatial structures delimited by the functional approach at the regional scale in eastern Canada). The foothills of coastal mountains (Groulx, Kapatahkatnahiu, etc.) have SWE values ranging between 250 and 400 mm.

In zone E (Fig. 10E), the estimated SWE values on the axis of Tremblant (998 m) and Raoul Blanchard (1175 m) mountains correspond to the values found on the northern summits at latitude 50°. At a lower altitude, in the Outaouais and Témiscamingue regions, the values of the SWE are between 100 mm and 250 mm. The St. Lawrence lowlands and southern regions have very low snow accumulations (100-200 mm). Southern parts of Chatigny (585 m) and Kaoskiwonatinak (627 m) mountains have estimated values between 300 mm and 400 mm, compared to100-200 mm in the Saguenay River watershed. The role of altitude on snow accumulation is also observed in the north and centre of zone F. The peaks of these mountains (Notre-Dame and Chic-Choc) have SWE values ranging between 250 mm and 400 mm. In coastal marine lands, values of 250 mm to 400 mm are estimated. At the southern limit between zone F with E are the St. Lawrence lowlands, where the estimated snow values are near the low SWE value of zone E (Fig. 10 F).

Overall, at the regional scale, the results of spatialization of the average maximum annual SWE based on the delimited structures are similar to the relief on the study area (Fig. 11). Indeed, the landscape (Canadian Shield, Appalachian, etc.) and the mountains north of the study area have high snow accumulation values. Towards the lowlands and coastal areas, the estimated average accumulations are low throughout the study area. The map of the SWE average annual maximum based on the grouped delimited structures approach resembles the results of the snow cover simulations done in eastern Canada by the Canadian Regional Climate Model (CRCM) and the Global Environment Multiscale in Climate Model (GEMCLIM) (Roy, 2006, and Dorsaz, 2008) (Fig. 11). These models have identified, at the regional scale (appx. 55 km), the isoline of SWE values up to 300 mm in the Torngat Mountains at the north, and 250 mm northeast on the central axis. Spatialization according to SWE structures demonstrates the same results as the regional values provided by the CRCM or the GEMCLIM. The GEMCLIM model overestimated the SWE on the coast of Labrador when compared to the results based on delimited structures (SWE values were overestimated by 100 mm in Blanc-Sablon, 200 mm in Natashquan, and 318 mm in Sept-Îles). Mapping of the average maximum annual SWE according to the structures generated more accurate results than the isolines of the regional model. Furthermore, the mapping of annual maximum SWE results is consistent with snow classes determined by Sturm et al. (1995) and is nearest to those observed by Langlois et al. (2014) in their study of the SWE simulation using remote sensing and in situ data on the regional models.

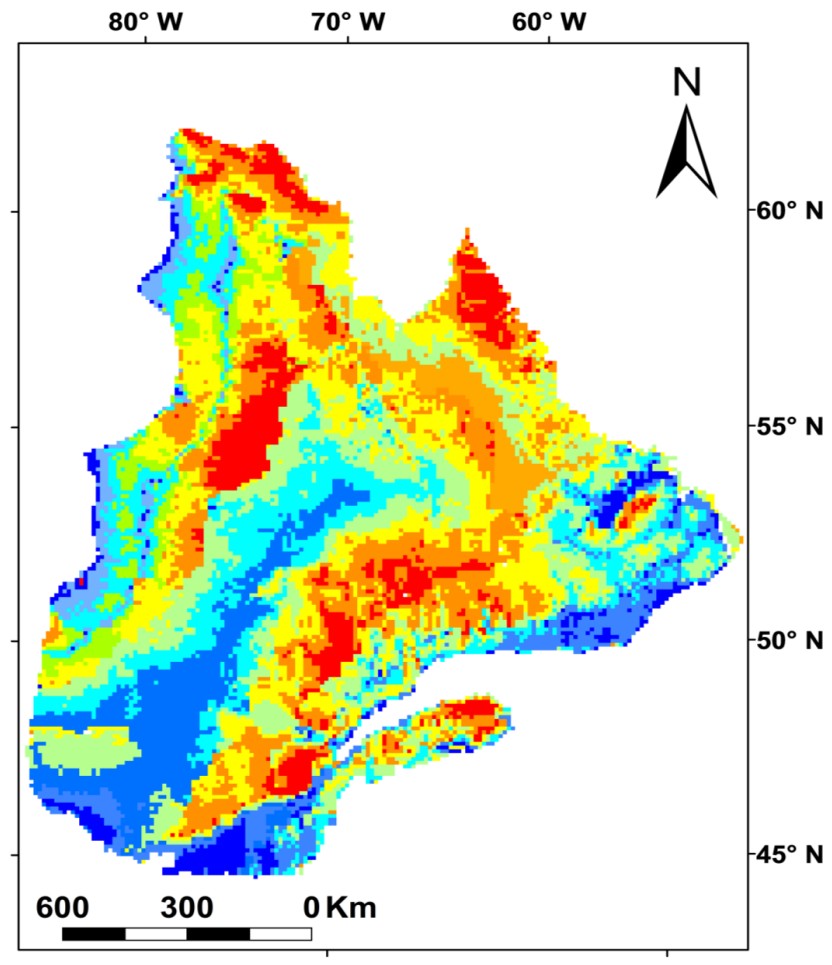

Figure 11: Average maximum annual SWE obtained by merging the spatial structures delimited at regional (10 km x10 km)

### 3.2.2 Local scale

20 At the local scale (300 m x 300 m), the snow variability is observed due to the slope, curvature forms, orientation of the watershed, vegetation height, etc. (Fig. 12). The results pin a specific value on the underlying processes taking place due to local physiographic factors. Indeed, north of zone A, the summits of the D'Youville mountains and foothills leading to the Canadian Shield have high SWE values (300-450 mm) (Fig. 12) but on the coast of Hudson and James bays, the high values of Qingaaluk (400 m) and Umiujaq (400 m) mountains is in contrast with the low value (250-300 mm) found along the bays.

25 Zone A is exposed to the dominant winds of the bays, where a low and medium accumulation (250-300 mm) is located in curvature forms. The forests located towards the hinterland are areas of high accumulations, with more than 300 mm.







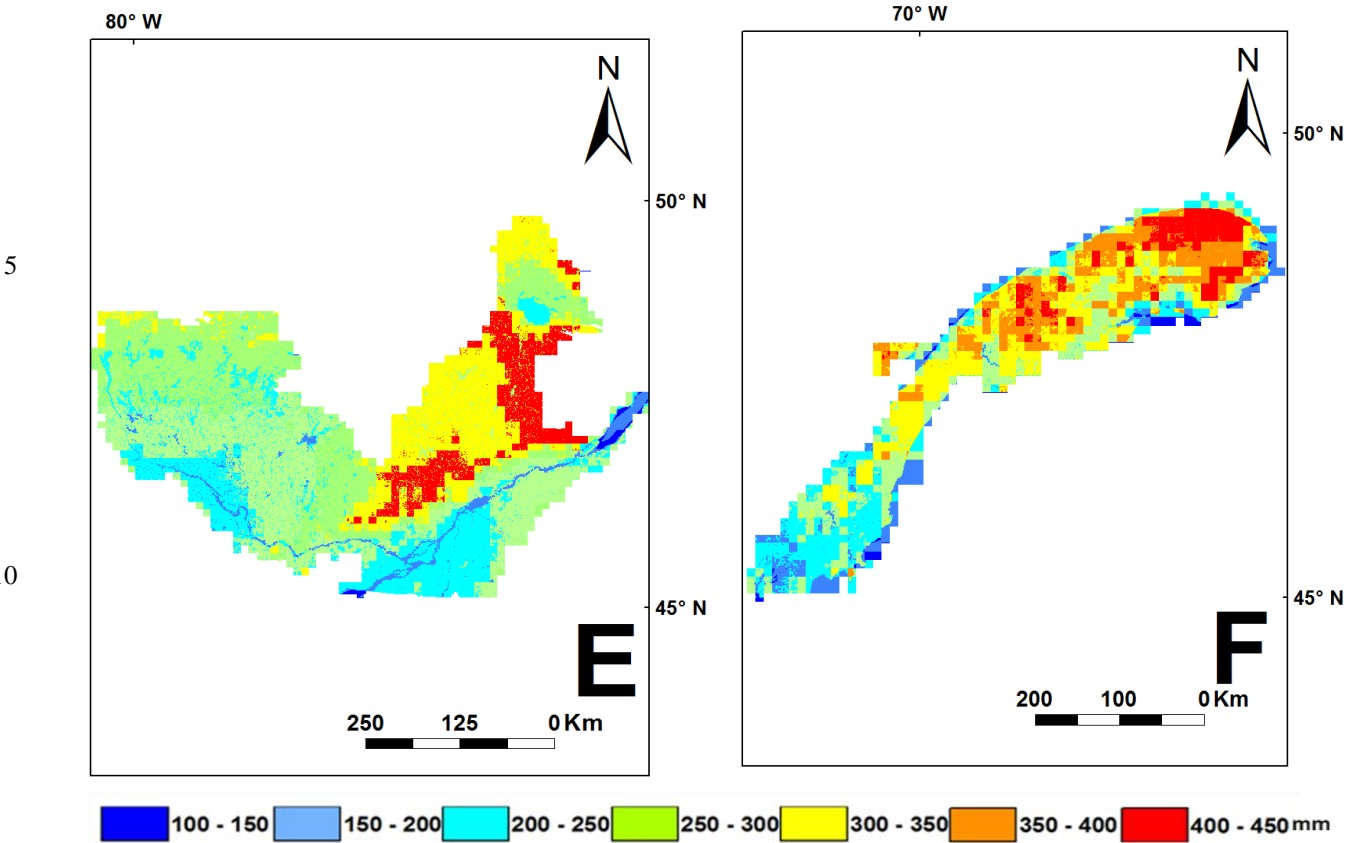

Figure 12: Average maximum annual SWE on the spatial structures delimited by the functional approach at the local scale (300 x 300 m) in eastern Canada

In zone B, the results confirm the accumulation identified by the model at the regional scale (Fig. 12B). At this observation scale, mountain lakes (Tichégami, Otish, Témiscamie, Severson and Uapahtkueh) have low values (200-250 mm), strongly contrasting with the high values (300-450 mm) of the mountains. These high SWE values are found on the sheltered slopes of the foothills of the northern mountains, which are exposed to humid ocean winds. The central part of zone B is a large series of lakes (Lake Caniapiscau at the north and Lake Mistassini in the southernmost centre). It is slightly hilly with SWE values ranging between 250 mm and 300 mm. To the west, the high values of SWE identified on Reid out (425 m) are intersected by the values (250-350 mm) of the western shield's foothills. Further south, the foothills of Chantigny mountains have SWE values ranging between 250 and 450 mm. The SWE of the southern part of zone B (the lowlands) ranges between 100 mm and 300 mm.

The estimated SWE in zone C (Fig. 12C) has lower average values (150-200 mm) into the means relief of Ungava Bay. However, this spatial estimation approach, based on the estimated resampling at the regional scale, does not model micro





reliefs (concavity and convexity forms) in the redistribution of the snow cover in this area. Hence, the summits swept by polar winds would presumably have very low SWE values, while the estimated values on watershed ubacs, which are moderately concave and not exposed to wind, would be high. In the far north, the valleys of the Pingualuit crater and coastal watershed would have high values and the hills and mountains would have lower values.

In zone D, which covers the mountains, foothills and lowlands east of the Otish and Groulx hills (Fig. 12D) the SWE spatialization returned high values (300-450 mm). These same values are found in the Kapatahkatnahiu hills, which constitutes a barrier to the humid oceanic winds. To the north and northwest, Happy Valley-Goose Bay mountains and valleys have an estimated SWE ranging between 250 mm and 350 mm. In the south, on either side of the Saguenay River, are the Grands Jardins National park (Raoul Blanchard, Conscrits etc.) and Valin hills (984 m), which had high SWE values (250-400 mm).

The areas located in topographic depressions (Manicouagan reservoir, St John, etc.) have low accumulations (0-150 mm). The lowlands of the northeastern coastline are high accumulation areas because the snow particles are trapped by forest formations.

The estimated values (100-250 mm) are prominent in zone E (Fig. 12E). The mountains found on the northeast (Sir Wilfried Laurier, Tremblant, Grands Jardins) have high estimated snow accumulation values, between 300 mm to 450 mm. Similarly, the Chantigny hills and foothills also have high SWE values (250 mm to 300 mm) and they intersect with the medium values of the Saguenay watershed (200-250 mm).

of the Saguenay watershed (200-250 mm). To the west, values of 200-250 mm are found over the uplands of Abitibi. The results show high values north of the Gatineau watershed and low values in the valleys (further south). This confirms the findings of Tapsoba et al. (2005) regarding the Gatineau watershed. SWE spatialization in zone F demonstrates the role of concave shapes in the accumulation of snow. In the Appalachian area, the SWE found in the concave shapes of the Chic-Chocs Mountains and valleys were estimated to be between 300 mm and 450 mm (Fig. 12F).

Overall, the estimated SWE values obtained at the local scale confirm the results found at the regional scale, with more variations in the structures (Fig. 13). The eastern part of the study area is influenced by the wet winds of Hudson Bay, the short vegetation, and the complex topographic curves and slopes, which explain the SWE average of 100-300 mm. On the foothills of the Canadian Shield, snow accumulations increase to 450 mm. The roles of the cooling air as a function of altitude and forests on the accumulation and retention of snow is observed on the eastern mountains (Groulx, Otish and Severson) and

summits of the Appalachian landscape, where snow values increase to more than 300 mm. The estimated values confirm the role played as a coastal barrier of the mountains, which block Atlantic wet winds. The centre of the study area has a low altitude and it is dominated by forests. To the south, the estimated values are generally low in the Laurentian lowlands and elevated in the highlands. This result confirms that the accumulation of snow is generally lower on the south of the study area than the north. In the north of the study area (Nunavik, for example), the sectors sheltered from polar winds have large accumulations

and the importance of micro-relief (curvature, slope, etc.) is once again demonstrated in the organization and maintenance of snow on the ground.



Figure 13: Average maximum annual SWE obtained by merging the spatial structures delimited at local scales (300 m x 300 m) in eastern Canada

## 4. Conclusion

This work is based on the explicit delineation of the spatial variability patterns of SWE conducted by Sena et al. (2015). This previous study by Sena et al. (2015) also addressed the regional and local scales. At the regional scale (10 km x 10 km), the study territory was divided into six geographical regions with homogeneous SWE spatial structures. These zones correspond to the major relief of the Canadian Shield (zones B, C, and D), the foothills of the shield (eastern part of zone A), the lowlands sectors (zone E) and the Appalachian relief (zone F). These spatial segmentation results are consistent with the divisions of major climate types, previous knowledge of major snow classes, and the limitations of snow density structures. Moreover, the



spatial segmentation results obtained provide precision on the limits of the spatial variability of SWE structures than those obtained by the Regional Climate Simulation Models (Langlois et al., 2014; Sturm et al., 1995). At the local scale (300 m x 300 m), these zones were segmented into small homogeneous SWE structures corresponding to the roles of slope morphology, vegetation height, slope, solar radiation and distance to lakes in accumulating and retaining snow on the ground.

The main objective of this work was to develop a multi-scale spatialization approach to estimate SWE which took into account the structures of the spatial variability of SWE at both regional and local scales (Sena et al., 2015). This goal was first achieved, at the regional scale, by applying the stepwise regression approach to regional physiographic metavariables $(U_1, U_2, U_3, U_4)$ and the average maximum annual SWE of all snow survey site. The residuals having a spatial structure as confirmed by a variogram analysis were retained to adjust the estimated values. At the local scale, snow accumulation and retention estimations

are a combination of regional estimated values and regional residual processing using local physiographic factors. In this case, the regional residuals for each area were modelled applying a stepwise regression on local physiographic metavariables. The estimated residual values allowed to adjust the estimate. All models were validated by a cross-validation method and confirmed by the index criteria (Nash, RSME, BIAIS). This method allows to take into account two components that are the deterministic and a part that is random (residual). In the deterministic part, the regression kriging model uses physiographic metavariables

as explanatory variables and in the random part, the regression residuals are calculated per region and undergo the variographic analysis process. Those with a spatial structure are kriged to account for the random component and improve the SWE estimate. At the regional scale (10 km), the stepwise regression model explains 68% of the variance. Only the residuals from zone E with a spatial structure were used to adjust the regional estimate. At the local scale (300 m), the stepwise regression models of the residuals improved the variance observed in the different zones. Overall, the adjustment of the regional SWE estimates

with the estimated local residuals explains a variance of 89%, improving the variance by 21% with a Nash value of 0.9. This shows that the SWE variability is explained by major regional variables (latitude, longitude, distance to the ocean and altitude) and approximately 21% of this variance is conditioned by local variables (slope, aspect, distance to rivers, solar radiation, curvature and vegetation height).

The map of the SWE was developed using PCI Geomatica and Arcgis softwares using metavariable physiographics on spatial

data forms. At the regional scale, the maps show the physiographic metavariables $(U_1, U_2, U_3, U_4)$ and kriged residuals of zone E on the spatial data forms. At the local scale, the maps of SWE was developed for each zone with the spatial metavariable physiographics of stepwise model of zone.

Two levels were distinguished from the results. Firstly, the SWE was estimated according to the physiographic factors which drive the dynamic structure and organize the accumulation of snow on the ground. The results of the SWE spatial estimation

show the role of elevation in snow accumulation at the regional scale. High mountains and highlands are opposed to moist winds loaded with snowfall. In each of the different geographic zones, high values of SWE are observed on mountain peaks and highlands. The effect of elevation is confirmed at the foothills of the Canadian Shield mountains. In zone A, on the western lowlands, high values are observed at the foothills of the shield. To the north, the Torngat, Pingualuit Crater and D'Youville mountains have heavy snow accumulations with estimated values between 300 mm and 450 mm (zone C). Similarly in the



Maritimes, the top of the Appalachian relief is an area of high accumulation (300 mm to 400 mm) (zone F). At the local scale, high values of SWE are observed on the concave forms of watersheds. Relief tops show SWE discontinuities accumulation corresponding to depressions or convex slope areas in the foothills. At this scale, SWE maps demonstrate the roles of physiographic variables (concave slope curvature, adret, etc.) in snow accumulation. The ubacs show high accumulation values

along the major mountains (Appalachians, Mont Groulx, etc.). On the mountains located in the eastern part of the study area, high values of SWE are observed on the high peaks forested. In the Maritimes, the eroded summits of the Appalachian relief are snow traps where the estimated SWE is over 300 mm. In the western part of the study area (zone A), the complexity of watershed shapes and low vegetation formations condition the estimated mean SWE values. Secondly, this approach improves the spatialization of the physical parameter of snow and provides data representative of the spatial variability of snow where

the snow survey network is not available. Also, this approach can be used for other purposes, such as determining the structure of the Climate Regional Model simulated data as the Canadian Regional Climate Model, for example. This was not the aim of the study.

This study is limited by the small number of snow survey stations (<4) in some areas (zone C, for example). Second, the availability, quality and accuracy of physiographic data at the desired resolution is also a limitation when applying the method

to other natural phenomena. This limit can influence the spatialization results in some areas and does not allow to adopt the approach at a larger scale.

This work introduces a thematic contribution to the new way of understanding SWE spatial variability by proposing SWE maps that take into account the limits of the structure of the SWE spatial variability at the regional (10 km x 10 km) and local (300 m x 300 m) scales. Other thematic contribution of this work includes the quantification of the percentage of the spatial

variability of the SWE at the local (89%) versus regional (68%) scales according to the corresponding physiographic variables. The adapted methodology and the results of this work offer several perspectives that can contribute to the study of the spatial variability of snow in a context of climate change. This method can be applied to other physical parameters of snow (density, height) and other variables of interest (annual minimum, monthly maximum, monthly average, etc.).

**Acknowledgements**

The authors thank the Ministère du Développement Durable, de l'Environnement de la Lutte contre les Changements Climatiques, Environment Canada, Rio Tinto and Hydro-Québec for providing the snow water equivalent data. The authors acknowledge assistance provided by Jessika Pickford for the English translation

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



## Appendices

### Appendix A.

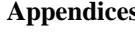

**Figure A1: SWE residue variogram analysis of areas B, D, and F at the regional scale**







**Figure A2: Estimated residus in zones A, B, D, E and F at the local scale**




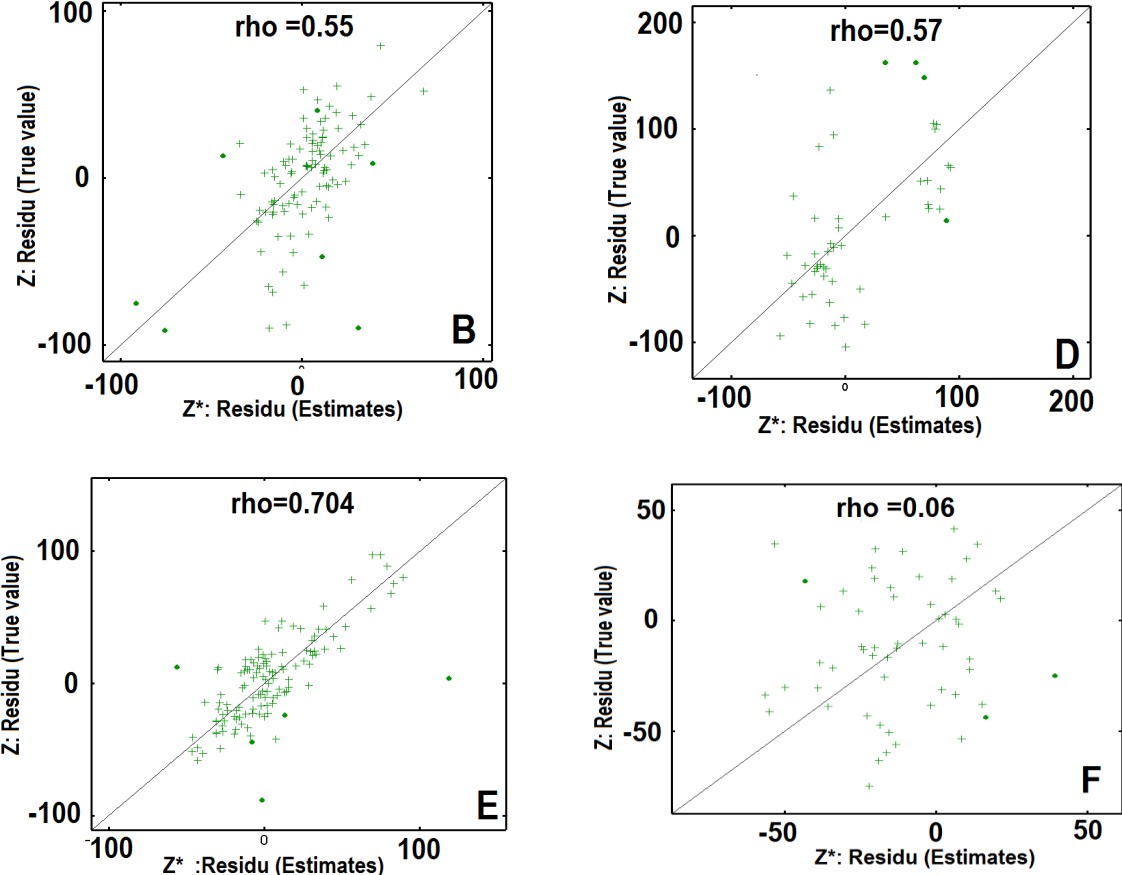

Figure A 3: Cross validation analysis on residuals of zones B, D, E, and F at the regional scale