# Peer review of "Multi-scale spatialization of snow water equivalent (SWE) according to their spatial structures in eastern Canada"

_The Cryosphere, 2020_

## Short Comment (SC1) · 22 Dec 2020

In Sena et al (in review) the authors introduce a spatially explicit estimate of snow water equivalent across northeastern Québec and Labrador. This reconstruction may be useful but I do have some questions about the data used to construct the model applied in this study. I would also like to call attention to there being significant overlap in some sections with an earlier manuscript (Sena et al [2019]) which includes whole sections of the snow data description being nearly word-for-word replicated from the earlier paper. This issue is quite glaring and surprising to see in a manuscript under consideration.

[Figure]

Some data related points are mentioned below: [1] In situ data is not summarized or shown on a map so it is difficult to surmise which regions are represented well. Likewise, it is difficult to know whether elevation bands, ecotypes and climate regions (coastal vs continental) are sampled appropriately by the data. The authors in Sena et al (2019) do present a map and it reveals huge swaths of land, particularly in eastern Labrador which have no SWE measurements included. This includes the Torngat Mountains where the authors are extrapolating from sea level to 1600+ m a.s.l. with no SWE observations within hundreds of kilometres of these sites. This is highly problematic to include these regions in the study in the absence of validation.

[2] The authors do not present their prediction errors which undoubtedly will be tremendously large in the areas that lack contemporary snow information. I find it difficult to see how this product is an improvement over reanalysis in many of the areas lacking snow survey information;

[3] Significant inter annual variability in snow cover occurred over the past 20 years so the authors need to test the assumption that this is not introducing extra error into the predictions when they are grouping together data with different periods. There are some stations on the map from Sena et al (2019) that have not been active for decades...;

[4] I am not presently sure I understand how the authors are determining SWE from weather stations that are only currently recording snow depth?

As a final note. The authors seemingly mention a lot of place names in Québec while largely avoiding the same for Labrador. I found the place names were a bit overwhelming overall, especially in the absence of a reference map. As such, consistency would be desired.

Sena, Chokmani, Gloaguen, and Bernier. 2019. Critical Analysis of the Snow Survey Network According to the Spatial Variability of Snow Water Equivalent (SWE) on Eastern Mainland Canada. Hydrology, 6: 55. doi:10.3390/hydrology6020055.

---

## Referee Comment (RC1) · Rhae Sung Kim (Referee) · 15 Jan 2021

The contribution of this work is to spatialize the average maximum SWE in eastern Canada at both (10km and 300m) scales according to spatial variability structure, which Sena et al. delineated in their previous study (Sena et al. 2015). This work is maybe useful, but the data description/validation is unclear and may be problematic. Also, the line numbering is not fully available in the manuscript, so it was difficult to leave comments. My comments focus on clarifying the methodology. - Snow data: Where is the location of the measurement sites (map)? How many samples for each site? What's the possible uncertainties and error of these measurements? Also, there is no

description of the measurement site. It's better to provide the maps of elevation, vegetation,.. of the study area for a better understanding of these sites. - Another thing that I am concerned with is the only use of the snow survey station data for both estimation and validation. Since the main trust of this paper is on SWE estimation and its validation, one would expect to see a more comprehensive probabilistic assessment of SWE estimation using a suite of measures to have a convincing analysis and conclusion. In addition, including regions of A and C is problematic because there are almost no observations there. - Analysis of regional/local physiographic factors is not fully or well explained. Is it possible to explain which factor is the dominant cause for each zone? Maybe an additional table including this information is helpful. - Section 3.2: It would be difficult to read this section for those who are not familiar with this region. The authors used a bunch of different names for stations, landscape, mountains ... without any locational information in the text. Quantified comparison results between estimated SWE maps and CRCM, GEMCLIM, Strum et al. (1995), and Langlois et al.(2014) are needed rather than just say "resemble" or "consistent". - Fig 8: three sub-figures are identical.

Specific comments:

1. What are the criteria for choosing both scales (10km vs 300m)?

2. Page 2, Line 11: remove "The spatial variability of the snow cover is explained by physiographic factors, which generate spatial structures at different scales."

3. Page 4, line 4: (MDDELCC, 2001) -> What means of 'MDDELCC'? I also had a hard time finding this citation in the reference.

4. Section 2.3: Which resampling method is used for SWE estimates at a local scale?

I stop here because the line numbering is not available after page 4.

---

## Author Comment (AC1) · 1 Feb 2021

Dear reviewer, I would like to thank you for the good reviews you made for this article. I have answered each of your questions and contributed relevant and essential answers for the understanding of the article. Question 1. In Sena et al (in review) the authors introduce a spatially explicit estimate of snow water equivalent across northeastern Québec and Labrador. This reconstruction may be useful but I do have some questions about the data used to construct the model applied in this study. I would also like to call attention to there being significant overlap in some sections with an earlier manuscript (Sena et al [2019]) which includes whole sections of the snow data description being

nearly wordfor-word replicated from the earlier paper. This issue is quite glaring and surprising to see in a manuscript under consideration. Answer1 The paragraph entitled "Snow data describing the snow water equivalent data" has been corrected (page 5 and 6, line 23-25) Question 2 In situ data is not summarized or shown on a map so it is difficult to surmise which regions are represented well. Likewise, it is difficult to know whether elevation bands, ecotypes and climate regions (coastal vs continental) are sampled appropriately by the data. The authors in Sena et al (2019) do present a map and it reveals huge swaths of land, particularly in eastern Labrador which have no SWE measurements included. This includes the Torngat Mountains where the authors are extrapolating from sea level to 1600+ m a.s.l. with no SWE observations within hundreds of kilometres of these sites. This is highly problematic to include these regions in the study in the absence of validation. Answer2 The in-situ data sites used in this study were added to Figure 1. In eastern Labrador, where the number of sites for measuring snow water equivalent is low (3 stations in total), the methodology adopted based on variographic analysis cannot be applied neither at the regional scale (10 km x 10 km) nor at the local scale (300 m x 300m) (see Figure 2). In the first part, at the regional scale, a multi-polynomial regression is developed as a function of the metavariables (U1,U2 ,U3 ,U4) and the mean annual maximum of the SWE of the all stations over the entire study area to estimate the snow water equivalent (for more details on the physiographic metavariables (for more informations, consulted Sena et al, 2015). At the local scale (300 m x 300 m), the resampling method was applied to the estimate obtained at the regional scale (10 km x 10 km). Question 3 The authors do not present their prediction errors which undoubtedly will be tremendously large in the areas that lack contemporary snow information. I find it difficult to see how this product is an improvement over reanalysis in many of the areas lacking snow survey information. Answer3. In paragraph 2.3.2 Local scale (page 8 line 13 -15 ), the statistical evaluation indices used, such as (determination coefficient (R2), BIAIS, relative mean squared error (RMSE), and Nash-Sutcliffe efficiency) are presented in Table 1, and thus the prediction errors of all models, all delineated zones explained at the regional and local

scales are presented in each figure. The same applies to the prediction error of the general model at the regional scale (Fig. 3) and at the local scale (Fig. 9). Question 4 Significant inter annual variability in snow cover occurred over the past 20 years so the authors need to test the assumption that this is not introducing extra error into the predictions when they are grouping together data with different periods. There are some stations on the map from Sena et al (2019) that have not been active for decades. Answer4 As mentioned in Section 2.2, of all historical data from snow stations in the study area, only those stations with an observation period of ten years or more are included in the study. For this study, we have assumed that the snow phenomenon is stationary during the observation period of ten years or more. Taking into account the interannual variability would be another challenge, which can be the subject of a later study taking into account this interannual variability of observation. This choice is not retained in this work. Question 5 I am not presently sure I understand how the authors are determining SWE from weather stations that are only currently recording snow depth? As a final note. The authors seemingly mention a lot of place names in Québec while largely avoiding the same for Labrador. I found the place names were a bit overwhelming overall, especially in the absence of a reference map. As such, consistency would be desired. Answer5 As mentioned in the snow data paragraph 2.2 , the study focuses on the water equivalent of snow, which is a physical parameter of snow cover. For this purpose, only the snow surveys of the study area were taken into account. It is important to remember that the snow survey stations measure only the physical parameters of the snow cover, which are: snow depth, snow density and snow equivalent on the ground. Reference names have been added

---

## Author Comment (AC2) · 1 Feb 2021

Dear reviewer, I would like to thank you for the good reviews you made for this article. I have answered each of your questions and contributed relevant and essential answers for the understanding of the article. Question 1 The contribution of this work is to spatialize the average maximum SWE in eastern Canada at both (10km and 300m) scales according to spatial variability structure, which Sena et al. delineated in their previous study (Sena et al. 2015). This work is maybe useful, but the data description/validation is unclear and may be problematic. Also, the line numbering is not fully available in the manuscript, so it was difficult to leave comments. My comments focus

on clarifying the methodology. – Snow data: Where is the location of the measurement sites (map)? How many samples for each site? What's the possible uncertainties and error of these measurements? Also, there is no description of the measurement site. It's better to provide the maps of elevation, vegetation, of the study area for a better understanding of these sites. - Another thing that I am concerned with is the only use of the snow survey station data for both estimation and validation. Since the main trust of this paper is on SWE estimation and its validation, one would expect to see a more comprehensive probabilistic assessment of SWE estimation using a suite of measures to have a convincing analysis and conclusion. In addition, including regions of A and C is problematic because there are almost no observations there. - Analysis of regional/local physiographic factors is not fully or well explained. Is it possible to explain which factor is the dominant cause for each zone? Maybe an additional table including this information is helpful. - Section 3.2: It would be difficult to read this section for those who are not familiar with this region. The authors used a bunch of different names for stations, landscape, mountains ... without any locational information in the text. Quantified comparison results between estimated SWE maps and CRCM, GEM-CLIM, Strum et al. (1995), and Langlois et al.(2014) are needed rather than just say "resemble" or "consistent". –

ANSWERS Several questions are included in this paragraph. Each has been taken separately to provide clarification. âĲŞ This work is maybe useful, but the data description/validation is unclear and may be problematic. Also, the line numbering is not fully available in the manuscript, so it was difficult to leave comments. Answer The description of the data used is presented in 2.2. The error of the models in each of the areas delineated at each observation scale is presented in all figures. The line numbering is corrected in the manuscript.

âĲŞ Snow data: Where is the location of the measurement sites (map)? How many samples for each site? What's the possible uncertainties and error of these measurements? Answer The distribution of snow survey stations is added to Figure 1. As noted

in Section 2.2, only stations with more than 10 years of observations are included. A critical review of the stations was carried out in Sena et al.2019. Errors in models developed in explicitly delineated areas are shown in each of the figures at both the regional and local scales.

âIJŞ Also, there is no description of the measurement site. It's better to provide the maps of elevation, vegetation,. of the study area for a better understanding of these sites. Answer This manuscript focuses on the spatialization of the snow water equivalent according to the limits of the different spatial structures delimited at the regional and local scale. The description of the snow survey stations and the different physiographic variables that condition the spatial variability of snow cover are further discussed in previous work by SENA et al, 2015 and 2019.

âIJŞ Another thing that I am concerned with is the only use of the snow survey station data for both estimation and validation. Since the main trust of this paper is on SWE estimation and its validation, one would expect to see a more comprehensive probabilistic assessment of SWE estimation using a suite of measures to have a convincing analysis and conclusion. Answer The method adopted takes into account the size of the data available in each of the zones with homogeneous spatial structures in terms of snow water equivalent (Séna et al.2015). The estimation of the SWE is a function of the physiographic meta-variables at the regional scale and of all the station data followed by variographic analysis of the clean residuals for each zone except Zone C (Fig.2 regional scale). At the local scale, it is the local residuals that are inputs to the models and that are combined with the estimates obtained at the regional scale. In this approach and in relation to the reduced data size, the goal is to spatialize the SWE as a function of the degree of variability conditioned by the different physiographic variables at each scale of observation. The probabilistic approach was not used in this study.

âIJŞ In addition, including regions of A and C is problematic because there are almost no observations there. Answer The few stations survey available for the water equivalent of snow in Zone C (3 stations) was - limited in the application of the selected

method. However, the method of resampling the regional snow water equivalent values made it possible to suggest a local estimate. In Zone A, the number of stations (14) available made it possible to apply the methodology adopted.

âIJŞ Analysis of regional/local physiographic factors is not fully or well explained. Is it possible to explain which factor is the dominant cause for each zone? Maybe an additional table including this information is helpful. Answer The approach adopted in this manuscript takes into account the physiographic metavariables U1, U2 ,U3 ,U4 obtained at the regional scale and others at the local scale U1LZ, U2LZ, U3LZ, U4LZ, U5LZ, U6LZ. The analysis of physiographic metavariables is carried out in the previous work of Sena et al.2015.

âIJŞ Section 3.2: It would be difficult to read this section for those who are not familiar with this region. The authors used a bunch of different names for stations, landscape, mountains ... without any locational information in the text. Answer Mountain and landscape names have been added to the maps.

âIJŞ Quantified comparison results between estimated SWE maps and CRCM, GEM-CLIM, Strum et al. (1995), and Langlois et al.(2014) are needed rather than just say "resemble" or "consistent". – Answer5 Corrections have been made.( line28-32, page 21)

Question 2 Fig 8: three sub-figures are identical. Specific comments: Answer2 Fig. 8 is corrected.

Question 3 What are the criteria for choosing both scales (10km vs 300m)? Answer3 The scale of observation of the phenomenon must be chosen taking into account previous studies and sufficiently large to cover the entire spatial variability of the phenomenon (Gustafson, 1998). In this study, the spatial variability of the SWE can only be measured at the scale that gives the spatial dimension of the data.At the local scale, the spatial variability of the physical parameters of the snow is measured on a 300m line of snow (MDDEFP 2008). At this observation scale, local variability is under the influence of specific local underlying processes. The regional observation scale selected is between 10 and 100 km and corresponds to the regional scale where the processes of the major atmospheric circulation agents are observed (Marsh 1999, McKay and Gray 1981). (For information, see Sena et al. 2015).

Question 4 Page 2, Line 11: remove "The spatial variability of the snow cover is explained by physiographic factors, which generate spatial structures at different scales." Answer4 The sentence is corrected.

Question 5 Page 4, line 4: (MDDELCC, 2001) -> What means of 'MDDELCC'? I also had a hard time finding this citation in the reference. Answer5 The reference is added in the bibliographic reference. The correction is made

Question 6 4. Section 2.3: Which resampling method is used for SWE estimates at a local scale? I stop here because the line numbering is not available after page 4. Answer6 Only the resampling of the estimated snow water equivalent value was carried out in Zone C. The resampling tool of the PCI Geomatica software was applied to the snow water equivalent values from the regional scale (10km x 10km) to the local scale (300m x 300m).

Reference Gustafson EJ (1998) Quantifying Landscape Spatial Pattern: What Is the State of the Art? Ecosystems (1):143-156. Marsh P (1999) Snowcover formation and melt: recent advances and future prospects. Hydrological Processes 13(14-15):2117-2134. McKay GA & Gray DM (1981) Distribution of snow cover in Handbook of Snow. 153-190 p MDDELCC (2008) Manuel d'instructions à l'usage des observateurs en nivométrie, Québec (Ministère du Développement Durable, de l'Environnement de la Lutte contre les Changements Climatiques, Québec), p32.

---

## Referee Comment (RC2) · Anonymous Referee #2 · 26 Feb 2021

There are two major issues that are not technical issues, but presentation. First, the submission is clearly not written by a native English speaker and not edited. The second and third sentences of the abstract are identical and this type of repetition occurs throughout the manuscript (see page 5). Terms including spatialize, physiographic regional factors, ubacs, etc. are not English terms and not defined. The writing is extremely rough making the work almost impossible to understand. An example is the presentation of the study goal "this study proposes to spatialize the SWE according to the structures of spatial variability of SWE. The main objective of this study is to develop a multi-scale spatialization approach by taking into account the structures delineated in the spatial variability analysis of the SWE at both scales (local and regional) by Sena et

al. (2015). Second the manuscript depends heavily on an earlier manuscript (Sena et al. 2015). Sena et al. (2015) is written in French and thus is not readily accessible for reference the target audience. Adequate information is required for this manuscript to be an independent submission. For example, there is no map of the validation stations and the physiographic regional factors are never defined (instead generic variables are used).

It was difficult to perform a proper technical review due to the writing challenges and limited information provided in this manuscript. The following technical issues were identified.

1. SWE and snow covered area are used interchangeably and it is not clear which is which. 2. The snow validation data set should be rewritten. Lead with a mapped set of stations that were used (move the second paragraph to the first). What was the time period? Then describe the sampling methods and the differences across the various networks. 3. It is not clear what the 10x10 km scale and 300 x 300 m scales mean. In part of the manuscript, it appears that there might have been an interpolation from the station data to a grid. 4. Section 2.3 needs to be written. The figure is not a standalone figure and the variables need to be defined. 5. What are the physiographic metavariables (ðİŚĹ1,ðİŚĹ2,ðİŚĹ3,ðİŚĹ4) and how were they calculated? 6. It is not clear that a step-wise linear regression is appropriate. Sena et al. (2015) used non-parametric methods. It does not appear that data were held back for validation purposes. 7. Section 3.1.2 is not a validation of the results, but a summary of the variogram metrics. This summaries would be best provided in a Table. 8. In section 3.1.3, the stepwise linear regression does not provide information about the explanatory variables and the difference in model performance for equations (5) to (9) is not reported. 9. Section 3.14 figures could be condensed by putting figures b and c together. Again, create a table of results rather than writing out in tabular form. Figure 8a, b, and c are identical and appear to be the same as Figure 7. 10. Section 3.2 leads with "At the regional scale (10 km x 10 km), the SWE spatialization was performed in each of the delimited

structure." It is not clear what analysis was conducted. The entirety of section 3.2 both at the local and regional scale seem to describe how much SWE there is where without any support. There is a tremendous amount of granularity that does not seem to be supported in many of the regions. In region A there are either 8 or 18 stations (both numbers were given) and region C has three stations. 11. The conclusions clearly indicate that there are physical factors that drive these variations but they are never described in the body of the text. In the conclusion, it is suggested that the authors have insights to what those physical features are "At the local scale (300 m x 300 m), these zones were segmented into small homogeneous SWE structures corresponding to the roles of slope morphology, vegetation height, slope, solar radiation and distance to lakes in accumulating and retaining snow on the ground." This is extremely valuable and important as compared to the average annual maximum values. 12. The conclusions introduce new information. The finding that "The adapted methodology and the results of this work offer several perspectives that can contribute to the study of the spatial variability of snow in a context of climate change." is not correct because there is no metric that would change due to a changing climate.

13. Table 1 does not match the statistics that are reported. 14. Units are missing in the nugget and variance; axes on various figures (Figure 3) are not labeled. Scatterplots should have the same size on the x and y-axis. Figure 3 has 3 subfigures but only (a) and (b) are labeled.

Overall, the methods could not be fully evaluated due to the deficiencies with the discussion. The findings appear to show spatial variability that is not warranted by the analysis. There is not compelling evidence that the kriging improves the models. The concept of variations that are driven at two different scales is reasonable and worthy of exploring, but the resulting models needed to be validated and the physical drivers of those models need to be identified at both scales and differentiated between scale.

---

## Referee Comment (RC3) · Anonymous Referee #3 · 3 Mar 2021

SWE survey stations are usually scarcely distributed and the SWE data doesn't necessary represent well larger region when spatially interpolated. This manuscript attempts to do spatial SWE interpolation with taking into account geographical and physical factors. In my opinion this goal is important and it could result with a better method to retrieve SWE maps. However, there are some major issues with the manuscript.

The spelling and grammar should be checked with a professional since there are obvious issues. For example, the second sentence in the abstract is repeated.

In addition, the structure of the manuscript is not coherent. Especially the Results-chapter (chapter 3) is hard to follow. In my opinion the chapter should be rethought

maybe by summarizing the results in a table rather than as text. The figures 4-8 contain lots of similar looking scatterplots. Is this really necessary? The key parameters could be in a single table without the plots.

Better maps (remaking Fig. 1 and 10) would improve the manuscript significantly. Since the manuscript deals with the spatial distribution of the snow surveys it would make sense to have a map of their locations. Even a geographic map of the target area would make following the discussion a lot easier since the text relies much on toponyms.

-Figure 1 should explain A-G in a) and colors in b) in the caption

-Chapter 2.2 Snow data, where are the stations located? Since the spatial distribution of stations is important there should be a map.

-Chapter 2.3.1 the metavariables U1-U4 are not explained. Are they same as latitude, longitude, altitude and distance to ocean? Or latitude and longitude, relief, and distance from the ocean?

-Chapter 2.4 the metavariables U1LZ,U2LZ,U3LZ,U4LZ,U5LZ,U6LZ are not explained. Are they same as slope, aspect, distance to rivers, solar radiation, curvature,and vegetation height?

-In chapter 3.1.1 and 3.1.2 the presentation of the results is hard to follow. Two tables would summarize the results better.

-Figures 4-8: is it really necessary to present multiple scatterplots for all zones? I think these results could be better summarized as a table.

-Figure 10: A-G should be explained in the caption

-The discussion in chapter 3.2 relies heavily on toponyms that the reader can't associate with the target region at all because the figures 1 and 10 are inadequate in this respect.

---

## Author Comment (AC5) · 30 Mar 2021

Dear reviewer, I would like to thank you for the good reviews and comments you made to improve the article. I have provided each of your pertinent questions with essential answers for the understanding of the article. These questions have generated modifications and additions of ideas that you have suggested. Thank you for your contribution.

Question1 There are two major issues that are not technical issues, but presentation. First, the submission is clearly not written by a native English speaker and not edited. The second and third sentences of the abstract are identical and this type of repetition
occurs throughout the manuscript (see page 5).

Answer1 The paper entitled Multi-scale spatialization of snow water equivalent (SWE) according to their spatial structures in eastern Canada is reviewed by the experts of Catalytic Translation (http://www.traductioncatalytik.com/) in scientific paper revision. Corrections have been made in the abstract. On page 5 (line 23-31), the paragraph is corrected with more precision.

Question2 Terms including spatialize, physiographic regional factors, ubacs, etc. are not English terms and not defined. Answer2 The Âń physiographic regional factors Âż is never used in this document. We used physiographic factors, physiographic metavariables. Ubacs is the side of a mountain that is the least exposed to the sun, and therefore the coldest side (north-facing slope (page1 , line 28).

Question3 The writing is extremely rough making the work almost impossible to understand. An example is the presentation of the study goal "this study proposes to spatialize the SWE according to the structures of spatial variability of SWE. The main objective of this study is to develop a multi-scale spatialization approach by taking into account the structures delineated in the spatial variability analysis of the SWE at both scales (local and regional) by Sena et al. (2015). Second the manuscript depends heavily on an earlier manuscript (Sena et al. 2015). Sena et al. (2015) is written in French and thus is not readily accessible for reference the target audience. Adequate information is required for this manuscript to be an independent submission. For example, there is no map of the validation stations and the physiographic regional factors are never defined (instead generic variables are used).

Answer 3 The information on the environmental variables used at the scales (regional and local) in the previous work of Sena et al. (2015) has been recalled. The method used is mentioned (Page 3, line 4-20) to make the article independent. In section 2.1, the area summaries are presented (Page 5 line 1 -25). The map of the SWE measurement stations is added.

Question4 1. SWE and snow covered area are used interchangeably and it is not clear which is which Answer4 The Âń snow covered area Âż is never used in this document. We used snow cover, snow survey stations, snow accumulations.

Question5 The snow validation data set should be rewritten. Lead with a mapped set of stations that were used (move the second paragraph to the first). What was the time period? Then describe the sampling methods and the differences across the various networks

Answer5 The section 2.3 Snow data has been corrected. The stations for measuring the physical parameters of the snow have been displayed in Figure 1. The methods of sampling the snow water equivalent and the choice of stations are described.

Question6 It is not clear what the 10x10 km scale and 300 x 300 m scales mean. In part of the manuscript, it appears that there might have been an interpolation from the station data to a grid

Answer6 The scale of observation of the phenomenon must be chosen by taking into account previous studies and sufficiently large to cover the entire spatial variability of the phenomenon (Gustafson, 1998). In this study, the spatial variability of the SWE can only be measured at the scale that gives the spatial dimension of the data. At the local scale, the spatial variability of the physical parameters of the snow is measured on a 300 m line of snow (MDDEFP 2008). At this observation scale, local variability is under the influence of specific local underlying processes. The regional observation scale selected is between 10 and 100 km and corresponds to the regional scale where the processes of the major atmospheric circulation agents are observed (Marsh 1999, McKay and Gray 1981). (For information, see Sena et al. 2015).

Question7 Section 2.3 needs to be written. The figure is not a standalone figure and the variables need to be defined

Answer7 Section 2.3 explains the methodology adopted. Figure 2 shows the different

methods applied at the different scales of observation of the SWE. The variables used are described at each scale in the section (Page 7 line 1-3). In section 2.3.1, the variables are defined and further detailed in the results.

Question8   5.          What    are    the    physiographic    metavariables U_l1,U_l2,U_l3,U_l4,U_l5,U_l6) and how were they calculated?

Answer8 The manuscript entitled "Multi-scale spatialization of snow water equivalent (SWE) according to their spatial structures in eastern Canada" is based on the results of the work of Sena et al.2015.   Also, in the manuscript, we had provided details on the physiographic metavariables (page 3 line 1-19). The metavariables U_1 ãĂŰ,UãĂŮ_2,U_3 ãĂŰ,UãĂŮ_4 of the regional scale and U_l1,U_l2,U_l3,U_l4,U_l5,U_l6 of the local scale are variables obtained in the previous studies by Sena et al. (2015).

Question9 It is not clear that a step-wise linear regression is appropriate. Sena et al. (2015) used non-parametric methods. It does not appear that data were held back for validation purposes

Answer9 In this study, stepwise linear regression is justified because the goal is to estimate SWE based on explanatory environmental metavariables that coordinate the spatial variability of SWE. And to do this, they are introduced into the regression model step by step. These explanatory environmental metavariables are not identical at both of the regional and local scales. In Sena et al., (2015), the non -parametric Kruskal-Wallis approach used shows that the SWE values located in a bounded spatial structure is different from the next contiguous structure. These are two different methods in the two papers.

Question 10 Section 3.1.2 is not a validation of the results, but a summary of the variogram metrics. This summary would be best provided in a Table. 8

Answer10 Section 4.2 is changed to variogram and cross-validation analysis, as it explains the results of the variogram models and the cross-validation of the residuals under the variogram model. Table 2 of the indices of the residuals variograms and cross-validation rho of the zones presents the summaries.

Question11 In section 3.1.3, the stepwise linear regression does not provide information about the explanatory variables and the difference in model performance for equations (5) to (9) is not reported

Answer11 Some details on the metavariables used are given. Equations 5 to 9 represent the regression models of the zones (A, B, D, E, F) with the metavariables considered in each zone. In the analysis of each model and the corresponding figures, the performances of Table 1 that are Nash, R2, RMSE and Biais are presented and discussed in the text. The difference in the performance of the models is mentioned (page 20, line 1-6).

Question12 Section 3.14 figures could be condensed by putting figures b and c together. Again, create a table of results rather than writing out in tabular form. Figure 8a, b, and c are identical and appear to be the same as Figure 7.

Answer12 Tables (2, 3 and 4) of results are created to summarize the performance indices of the models. Figures 8a, b and c are corrected.

Question13 Section 3.2 leads with "At the regional scale (10 km x 10 km), the SWE spatialization was performed in each of the delimited structure." It is not clear what analysis was conducted. The entirety of section 3.2 both at the local and regional scale seem to describe how much SWE there is where without any support. There is a tremendous amount of granularity that does not seem to be supported in many of the regions. In region A there are either 8 or 18 stations (both numbers were given) and region C has three stations

Answer13 The mapping of the SWE to the delineated spatial structures is presented at the beginning of the paragraph (Page 22, line 3-4). In the methodology, information

about the data and the tool used is mentioned (page 9 and 10, line 20 -23 ). In area A, there are 18 stations and 3 stations in area C (Page 10, line 10-11).

Question14 The conclusions clearly indicate that there are physical factors that drive these variations but they are never described in the body of the text. In the conclusion, it is suggested that the authors have insights to what those physical features are "At the local scale (300 m x 300 m), these zones were segmented into small homogeneous SWE structures corresponding to the roles of slope morphology, vegetation height, slope, solar radiation and distance to lakes in accumulating and retaining snow on the ground." This is extremely valuable and important as compared to the average annual maximum values

Answer14 This manuscript derived from the results of the work of Sena et al.2015, which explicitly delineated the different structures of spatial variability at the regional and local scales. The physical factors guiding the spatial variability of SWE are described in the work of Sena et al.(2015) and their roles studied in explicitly delineating the structures of spatial variability of SWE. Meta-variables are obtained from these variables. In this work, the continuous mapping of SWE values in each structure is done by considering the contained data of each zone. This work allowed us to estimate the role of major factors such as altitude, longitude, latitude and distance from the ocean in the spatial variability of snow cover at 68%. To this is added the 21% variance of local factors (slope, distance to lakes etc.) in coordinating the spatial variability of snow cover on the ground.

Question15 The conclusions introduce new information. The finding that "The adapted methodology and the results of this work offer several perspectives that can contribute to the study of the spatial variability of snow in a context of climate change." is not correct because there is no metric that would change due to a changing climate.

Answer15 The spatialisation in this study is based on the homogeneous spatial structures in terms of the SWE. These structures are different from each other at both scales

(regional and local) (Sena., 2015). In case we will have simulated data of climate variables (density, height, EEN, rainfall) produced by the Canadian Regional Climate Model for example, or simulated data on the evolution of vegetation formations (plant heights) in context of climate change, the limit of structures can be modified or changed. The method can be used to compare the evolution of spatial structures with the spatial variability of the natural phenomenon in a future climate (Page 33, line 27-31). This may induce a change in the spatialisation of the physical parameters of the snow cover.

Question16 Table 1 does not match the statistics that are reported.

Answer16 In all model figures the statistic indices in Table 1 are presented and discussed. Question17 Units are missing in the nugget and variance; axes on various figures (Figure 3) are not labeled. Scatterplots should have the same size on the x and y-axis. Figure 3 has 3 subfigures but only (a) and (b) are labeled.

Answer17 Units of variance and nugget effect are added (Figure 3). In Figure 3 the scatter plots have the same size. Figure 3 is corrected.

Question18 Overall, the methods could not be fully evaluated due to the deficiencies with the discussion. The findings appear to show spatial variability that is not warranted by the analysis. There is not compelling evidence that the kriging improves the models. The concept of variations that are driven at two different scales is reasonable and worthy of exploring, but the resulting models needed to be validated and the physical drivers of those models need to be identified at both scales and differentiated between scale.

Answer18 The regression method has two components: the deterministic part and the random part (residuals). This random part is studied by the variogram to demonstrate the existing spatial structure. Those that show spatial structure are kriged to account for the random component and improve the SWE estimated. The use of regression kriging allows the estimation of the average annual maximum of the SWE at all points in the territory at the regional and local scales. The regression model explains 65% of

the variance (Nashr = 56%). With the addition of the kriged residuals of area E, this variance is reduced to 68% with a Nashr of 83%. The physical factors that affect the spatial variability of the snow cover (in this case the SWE) are not identical at both scales (regional and local). They have been identified, analyzed, and validated in the previous work of Sena et al.2015. These physical factors allowed to explicitly delineate the structures of the variability of the SWE. And this manuscript proposes to spatialise the SWE according to these structures.

Bibliography reference Gustafson EJ (1998) Quantifying Landscape Spatial Pattern: What Is the State of the Art? Ecosystems (1):143-156. Marsh P (1999) Snowcover formation and melt: recent advances and future prospects. Hydrological Processes 13(14-15):2117-2134. McKay GA & Gray DM (1981) Distribution of snow cover in Handbook of Snow. 153-190 p MDDELCC (2008) Manuel d'instructions à l'usage des observateurs en nivométrie, Québec (Ministère du Développement Durable, de l'Environnement de la Lutte contre les Changements Climatiques, Québec), 120p

---

## Author Comment (AC6) · 30 Mar 2021

I would like to thank you for the good reviews and comments you made to improve the article. I have provided each of your pertinent questions with essential answers for the understanding of the article. These questions have generated modifications and additions of ideas that you have suggested. Thank you for your contribution.

Question 1 SWE survey stations are usually scarcely distributed and the SWE data doesn't necessary represent well larger region when spatially interpolated. This manuscript attempts to do spatial SWE interpolation with taking into account geographical and physical factors. In my opinion this goal is important and it could result with a better method to retrieve SWE maps. However, there are some major issues with the manuscript. The spelling and grammar should be checked with a professional since there are obvious issues. For example, the second sentence in the abstract is repeated.

Answer 1 The paper entitled Multi-scale spatialization of snow water equivalent (SWE) according to their spatial structures in eastern Canada is reviewed by the experts of Catalytic Translation (http://www.traductioncatalytik.com/) in scientific paper revision. Corrections have been made in the abstract. On page 5 (line 23-31), the paragraph is corrected with more precision.

Question 2 In addition, the structure of the manuscript is not coherent. Especially the Results-chapter (chapter 3) is hard to follow. In my opinion the chapter should be rethought maybe by summarizing the results in a table rather than as text. The figures 4-8 contain lots of similar looking scatterplots. Is this really necessary? The key parameters could be in a single table without the plots.

Answer 2 Figures 4 to 8 represent results from the different models in the delineated areas. Summary tables (3 in total) have been added to simplify the presentation of the results. These scatterplots allow to visualize the organization of the points according to the 1:1 diagonal.

Question 3

Better maps (remaking Fig. 1 and 10) would improve the manuscript significantly. Since the manuscript deals with the spatial distribution of the snow surveys it would make sense to have a map of their locations. Even a geographic map of the target area would make following the discussion a lot easier since the text relies much on toponyms.

Answer 3 The map of the distribution of measurement stations is added to Figure 1. The names of the largest mountains and some snow measuring stations have been

added to locate the reader.

Question 4 -Figure 1 should explain A-G in a) and colors in b) in the caption

Answer 4 Explanations of the different zones delineated at the regional scale are provided (Page5, line 1-16). Explanations of Figure 1b are added. (Page5, line 17-21)

Question 5 -Chapter 2.2 Snow data, where are the stations located? Since the spatial distribution of stations is important there should be a map.

Answer 5 The location of the stations is added to Figure 1

Question 6 -Chapter 2.3.1 the metavariables U1-U4 are not explained. Are they same as latitude, longitude, altitude and distance to ocean? Or latitude and longitude, relief, and distance from the ocean?

Answer 6 Metavariables U1 to U4 are explained (Page 3, line 1-21)

Question 7 -Chapter 2.4 the metavariables U1LZ,U2LZ,U3LZ,U4LZ,U5LZ,U6LZ are not explained. Are they same as slope, aspect, distance to rivers, solar radiation, curvature,and vegetation height?

Answer 7 Metavariables U1LZ to U6LZ are explained (Page 3, line 1-21)

Question 8 -In chapter 3.1.1 and 3.1.2 the presentation of the results is hard to follow. Two tables would summarize the results better.

Answer 8 Tables (3 in total) have been added to make a summary of the results.

Question 9

-Figures 4-8: is it really necessary to present multiple scatterplots for all zones? I think these results could be better summarized as a table.

Answer 9 The point scatterplots of the different models are different from one area to another. The presentation allows to visualize the distribution of the points with respect to the 1;1 diagonal. The results of the indices are presented in tables

Question 10 -Figure 10: A-G should be explained in the caption

Answer 10 Figure 10 shows the delineated areas presented (Page 4, Figure 1a) with the map of the SWE estimated.

Question 11

-The discussion in chapter 3.2 relies heavily on toponyms that the reader can't associate with the target region at all because the figures 1 and 10 are inadequate in this respect.

Answer 11 Toponyms are added to the maps.